

# The Tipping Points Modelling Intercomparison Project (TIPMIP): Assessing tipping point risks in the Earth system

Ricarda Winkelmann[1,2,3*], Donovan P. Dennis[1,2], Jonathan F. Donges[1,2,4], Sina Loriani[1,2], Ann Kristin Klose[2,3], Jesse F. Abrams[5], Jorge Alvarez-Solas[6,7], Torsten Albrecht[1,2], David Armstrong McKay[5,8], Sebastian Bathiany[9,10], Javier Blasco Navarro[11], Victor Brovkin[12,13], Eleanor Burke[13], Gokhan Danabasoglu[14], Reik V. Donner[10,15], Markus Drüke[10,16], Goran Georgievski[12], Heiko Goelzer[17], Anna B. Harper[18], Gabi Hegerl[19], Marina Hirota[20], Aixue Hu[14], Laura C. Jackson[13], Colin Jones[21], Hyungjun Kim[22], Torben Koenigk[23], Peter Lawrence[14], Timothy M. Lenton[5], Hannah Liddy[24,25], José Licón-Saláiz[2], Maxence Menthon[26], Marisa Montoya[6,7], Jan Nitzbon[11], Sophie Nowicki[27], Bette Otto-Bliesner[14], Francesco Pausata[28,29], Stefan Rahmstorf[10], Karoline Ramin[2], Alexander Robinson[11], Johan Rockström[2,10,30], Anastasia Romanou[25,31], Boris Sakschewski[10], Christina Schädel[32], Steven Sherwood[33,34], Robin S. Smith[35], Norman J. Steinert[36], Didier Swingedouw[37], Matteo Willeit[10], Wilbert Weijer[38,39], Richard Wood[13], Klaus Wyser[23], Shuting Yang[40]

*Correspondence to:* Ricarda Winkelmann (winkelmann@gea.mpg.de)

[1]Integrative Earth System Science, Max Planck Institute of Geoanthropology, Jena, Germany
[2]Earth Resilience Science Unit, Potsdam Institute for Climate Impact Research, Member of the Leibniz Association, Potsdam, Germany
[3]Institute of Physics and Astronomy, University of Potsdam, Potsdam, Germany
[4]Stockholm Resilience Centre, Stockholm University, Stockholm, Sweden
[5]Global Systems Institute, University of Exeter, Exeter, UK
[6]Institute of Geosciences, CSIC-UCM, Madrid, Spain
[7]Department of Earth Physics and Astrophysics, Complutense University of Madrid, Madrid, Spain
[8]School of Global Studies, University of Sussex, Brighton, UK
[9]Department of Earth System Modeling, School of Engineering and Design, Technical University of Munich, Munich, Germany
[10]Potsdam Institute for Climate Impact Research, Member of the Leibniz Association, Potsdam, Germany
[11]Alfred Wegener Institute Helmholtz Centre for Polar and Marine Research, Potsdam, Germany
[12]Max Planck Institute for Meteorology, Hamburg, Germany
[13]Met Office Hadley Centre, Exeter, UK
[14]US National Science Foundation, National Center for Atmospheric Research, Boulder, USA
[15]Magdeburg-Stendal University of Applied Sciences, Magdeburg, Germany
[16]Department of Hydrometeorology, Deutscher Wetterdienst, Germany
[17]NORCE Norwegian Research Centre, Bjerknes Centre for Climate Research, Bergen, Norway
[18]University of Georgia, Department of Geography, Athens, USA
[19]School of GeoSciences, University of Edinburgh, Edinburgh, UK
[20]Group IpES, Department of Physics, Federal University of Santa Catarina, Florianópolis, Brazil
[21]National Centre for Atmospheric Science, School of Earth and Environment, University of Leeds, Leeds, UK
[22]Korea Advanced Institute of Science and Technology, Daejeon, Korea
[23]Swedish Meteorological and Hydrological Institute, Norrköping, Sweden
[24]Columbia Climate School, Columbia University, New York, NY, USA
[25]NASA Goddard Institute for Space Studies, New York, NY, USA
[26]Vrije Universiteit Amsterdam, Earth Science Department, Amsterdam, The Netherlands
[27]Department of Earth Science, University at Buffalo, Buffalo, New York, USA



[28]ESCER (Centre pour l'étude et la simulation du climat à l'echelle régionale), Department of Earth and Atmospheric Sciences, University of Quebec in Montreal, Montreal, Canada
[29]GEOTOP (Research Center on the Dynamics of the Earth System), Department of Earth and Atmospheric Sciences, University of Quebec in Montreal, Montreal, Canada
[30]Institute of Environmental Science and Geography, University of Potsdam, Potsdam, Germany
[31]Department of Applied Physics and Applied Mathematics, Columbia University, New York, USA
[32]Woodwell Climate Research Center, Falmouth, MA, USA
[33]Climate Change Research Centre, University of New South Wales, Sydney, New South Wales, Australia
[34]ARC Centre of Excellence for Climate Extremes, University of New South Wales, Sydney, New South Wales, Australia
[35]National Centre for Atmospheric Science, Department of Meteorology, University of Reading, Reading, UK
[36]CICERO Center for International Climate Research, Oslo, Norway
[37]Environnements et Paléoenvironnements Océaniques et Continentaux (EPOC) Université de Bordeaux, CNRS, Bordeaux, France
[38]Los Alamos National Laboratory, Los Alamos, NM, USA
[39]International Arctic Research Center, University of Alaska Fairbanks, Fairbanks, AK, USA
[40]Danish Meteorological Institute, Copenhagen, Denmark

**Abstract.** The risk of transgressing critical thresholds triggering nonlinear change in the Earth system increases with rising human pressures from greenhouse gas emissions, land-use change and other drivers. Several key components of the Earth System such as the Greenland and Antarctic ice sheets, permafrost, the Atlantic Meridional Overturning Circulation (AMOC), and boreal and tropical forests as well as mountain glaciers, terrestrial hydrological systems and the Sahel region have been

suggested to exhibit self-amplifying feedback processes that could lead to non-linear and often abrupt and/or irreversible transitions with far-reaching biophysical and socio-economic consequences. While concerns within the scientific community, general public and among policy- and decision-makers are growing, significant uncertainties remain regarding the critical thresholds, timescales, interactions and impacts of such tipping dynamics. To address these critical knowledge gaps, we here present the Tipping Points Modelling Intercomparison Project (TIPMIP), an international collaborative effort to systematically

assess tipping point risks using state-of-the-art coupled Earth System Models and stand-alone domain models, e.g., ice sheet and land system models. Building on the success of established Model Intercomparison Projects (MIPs), TIPMIP aims to standardize model experiments, quantify uncertainty ranges for critical forcing thresholds, and provide a multi-model assessment of tipping dynamics across key Earth system components. TIPMIP will enhance our ability to anticipate and mitigate the risks associated with Earth system tipping points, and support science-based decision-making in the face of high-

end impacts and deep uncertainties.

## 1 Introduction

As the last year saw unprecedented temperatures, making 2024 the first calendar year with global warming exceeding 1.5°C above pre-industrial levels (Copernicus, 2025), every region on Earth is increasingly affected by climate change impacts in multiple ways, including more frequent and more severe extreme events, changes in precipitation patterns, and sea-level rise,

with far-reaching impacts on biodiversity loss, soil degradation, and water, energy and food securities (Masson-Delmotte et



al., 2021; Masson-Delmotte et al., 2021; WWF, 2024). With further warming, we are facing growing risks of large-scale non-linear shifts in the Earth system (Lee et al., 2021; IPCC, 2023). Among these are risks through tipping dynamics (Lenton et al., 2008; Armstrong McKay et al., 2022; Lenton et al., 2023), which are triggered when a change in part of the Earth System becomes self-perpetuating beyond a critical threshold (tipping point), which can lead to substantial, widespread, often abrupt

and/or irreversible biophysical impacts that can further induce far-reaching socioeconomic impacts (see Box 1).

Key components of the Earth system – including the Greenland and Antarctic ice sheets, permafrost, the Atlantic Meridional Overturning Circulation (AMOC), the boreal and tropical forests, among others – are associated with such self-amplifying feedbacks (e.g., Armstrong McKay et al., 2022; Lenton et al., 2023). Transgressing critical thresholds in these key components of the Earth system would threaten the health and livelihoods of billions of people (Defrance et al. 2017; Lenton et al., 2023).

While there is overall consensus on the possibility of nonlinear and irreversible responses of parts of the climate system within the scientific community (Lee et al., 2021), the detailed nature and characteristics of tipping dynamics constitute an active research frontier. In particular, it is essential to better define, disentangle and critically assess the underlying feedbacks and traits of abruptness and irreversibility on different temporal and spatial scales (Kopp et al., 2024; Stocker et al., 2024). Critical knowledge gaps remain, not least due to the very nature of non-linear phenomena, that give rise to large uncertainties and

possibly unexpected changes after a long period of perceived stability (Feudel, 2023). Further, some of the underlying feedbacks and/or interactions between different processes in the Earth system are not fully understood, partly missing and/or inadequately represented in models, and there is a lack of observational data for model input and validation, especially on longer timescales (Loriani et al., 2025).

Despite – or precisely because of – these uncertainties, risks associated with tipping dynamics are raised as a central issue by

policy- and decision-makers as well as scientific assessment bodies: The most recent Synthesis Report of the Intergovernmental Panel on Climate Change (IPCC, 2023), states that the "likelihood of abrupt and irreversible changes and their impacts increases with higher global warming levels (high confidence). [...] Risks associated with large-scale singular events or tipping points, such as ice sheet instability or ecosystem loss from tropical forests, transition to high risk between 1.5°C to 2.5°C (medium confidence) and to very high risk between 2.5°C to 4.0°C (low confidence)." This statement simultaneously

highlights the potential high risks posed by tipping dynamics, the substantial uncertainties associated with them, and the need for additional and more robust lines of evidence (Stocker et al., 2024). This is also reflected by the summary table IPCC-AR6 WG1 4.10 (Lee et al., 2021), which displays considerable uncertainties for most potential tipping systems under consideration. In scoping discussions for the upcoming IPCC 7th Assessment cycle, significant interest was expressed among the United Nations Framework Convention on Climate Change (UNFCCC) member states to shed more light on these uncertainties, with

the topic of tipping points receiving the highest endorsement for a special report (IPCC Secretariat, 2024). Although this proposal was ultimately not approved, a dedicated chapter on "Abrupt changes, low-likelihood high impact events and critical thresholds, including tipping points, in the Earth system" is planned for the IPCC-AR7 WG1 report.



This assessment follows in a sequence of earlier reports, in which the assessed risk of crossing tipping points in the Earth system at a given warming level has tended to increase over the past 25 years, while the need to further reduce the substantial

associated uncertainties has remained high (Fig. 1).

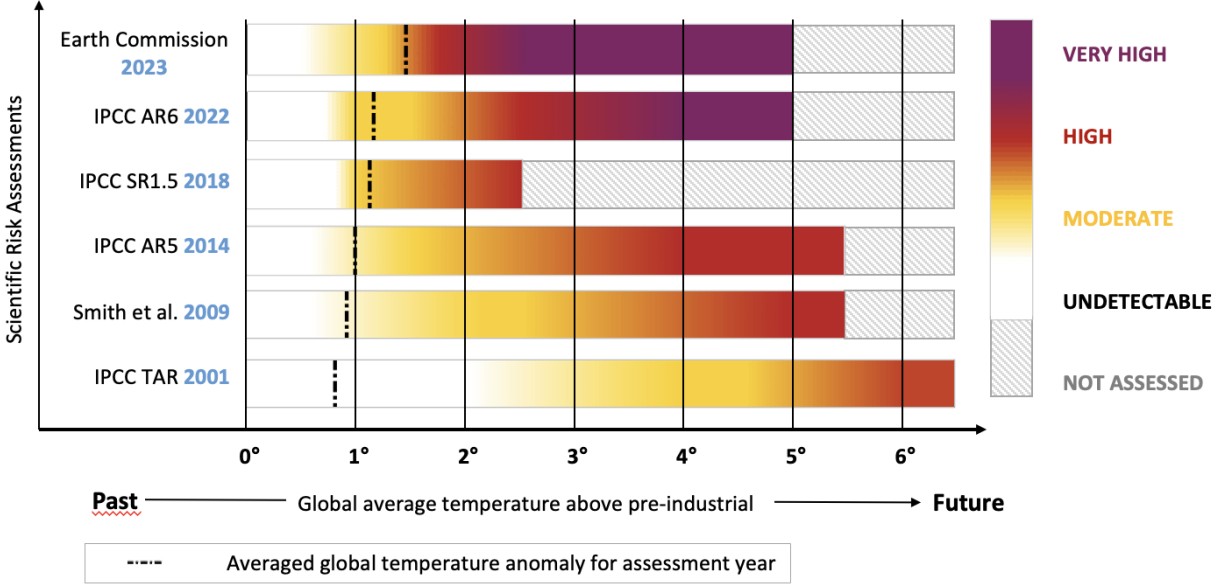

**Figure 1: Level of risk associated with large-scale singular events or tipping dynamics in scientific assessment reports**. Figure updated from Lenton et al. (2019), with quantifications as in Marbaix et al. (2024). A table summarizing the risk assessments which inform the figure is included in the supplementary materials as Appendix A.

The notion of tipping points has also been picked up by social movements, business leaders and indigenous rights advocates (OECD, 2022; Roman Cuesta et al., 2025; Laybourn et al., 2024; Zhong and Rojanasakul, 2024). One of the central issues discussed in this context is that of justice, as tipping dynamics would exacerbate the impacts of climate change, and those regions which have contributed the least to greenhouse gas emissions in the past are and will likely be among the most severely

affected by these impacts (Pörtner et al., 2022). Tipping dynamics could also change the local pattern and even sign of climate impacts (e.g., a substantial weakening of the AMOC could lead to a relative cooling over Northern Europe (Jackson et al., 2015).

Moreover, the long response times of certain tipping systems and their self-sustained nature, can lead to legacy effects. For example, warming in the coming years to decades could trigger tipping dynamics in the ice-sheets contributing to continued

sea-level rise for centuries to come (e.g., Golledge et al., 2015; Klose et al., 2024). Owing to such committed, time-delayed changes (see Box 1), it is clear that future generations would bear the bulk of the impacts caused by crossing potential tipping points.



Tipping dynamics thus touch upon critical issues of both international and intergenerational justice (Rockström et al., 2023; Gupta et al., 2023), and make it inherently very difficult to develop adequate response or adaptation measures. The COP28

Global Stocktake further highlights that "improved understanding of how to avoid and respond to the risk of low-likelihood or high-impact events or outcomes, such as abrupt changes and potential tipping points [...] are needed to comprehensively manage risks of and respond to loss and damage associated with climate change impacts". While it is clear that exceeding tipping points would result in "high impacts" (as suggested by paleoclimatic evidence and insights from numerical modelling), based on our current understanding tipping dynamics should in fact be considered "unknown likelihood, high impact" events

(i.e. the likelihood could be low or medium or high), with strong implications for risk governance (Milkoreit et al., 2024).

**In light of these far-reaching implications, we here propose the Tipping Points Modelling Intercomparison Project (TIPMIP), to address the critical knowledge gaps regarding tipping dynamics in an international community effort, and to work towards a multi-model assessment of the associated likelihoods and biophysical impacts.** As of yet, there is no such systematic approach to study tipping point risks in a standardized way across biophysical domains. Within TIPMIP,

we aim at providing such a systematic analysis for key Earth system components (see Fig. 2), including the ice-sheets, ocean, permafrost and different biomes, based on state-of-the-art Earth System Models as well as stand-alone domain models.

Numerical process-based modelling allows us to advance our understanding of crucial feedbacks, and to quantify uncertainty ranges for critical thresholds based on ensembles of scenarios, modelling choices and parameters. Multi-model assessments and Model Intercomparison Projects (MIPs) – e.g., CMIP (the Coupled Model Intercomparison Project; e.g., Eyring et al.,

2016), ISMIP (the Ice Sheet Model Intercomparison Project; e.g., Nowicki et al., 2020), ScenarioMIP (the Scenario Model Intercomparison Project; e.g., O'Neill et al., 2016), or ISIMIP (the Inter-Sectoral Impact Model Intercomparison Project; e.g., Frieler et al., 2017) – wherein standardized inputs allow for the evaluation of impacts across sectors and modelling domains, are well-established and widely used to improve and consolidate scientific understanding and thereby to support and inform climate policy. Bringing together results from multiple models, and further lines of evidence from paleoclimate records and

observations is particularly important in the context of tipping dynamics as we are entering into uncharted territory: Already today, human societies are confronted with climate impacts which are "unprecedented over many centuries to many thousands of years" (IPCC, 2023). Tipping dynamics would further amplify these unprecedented climate impacts, which are potentially irreversible and could lead to unacceptable risks.

Here we introduce the various levels of complexity and define different aspects of tipping dynamics assessed within TIPMIP

(Section 2), and present the current state of knowledge with respect to these aspects and characteristic traits with a focus on the Greenland and Antarctic ice sheets, AMOC, permafrost, boreal and tropical forests, and potential regional tipping systems (Section 3). We further detail the aims of TIPMIP and key research questions (Section 4). In Section 5, we describe the experimental design and how it will be applied across different tipping systems and different types of models, as well as the governance structure of TIPMIP. In Section 6, we give an overview of the methods to analyse the model simulations and

combine the results in an overarching risk assessment, followed by an outlook on future research activities in Section 7.



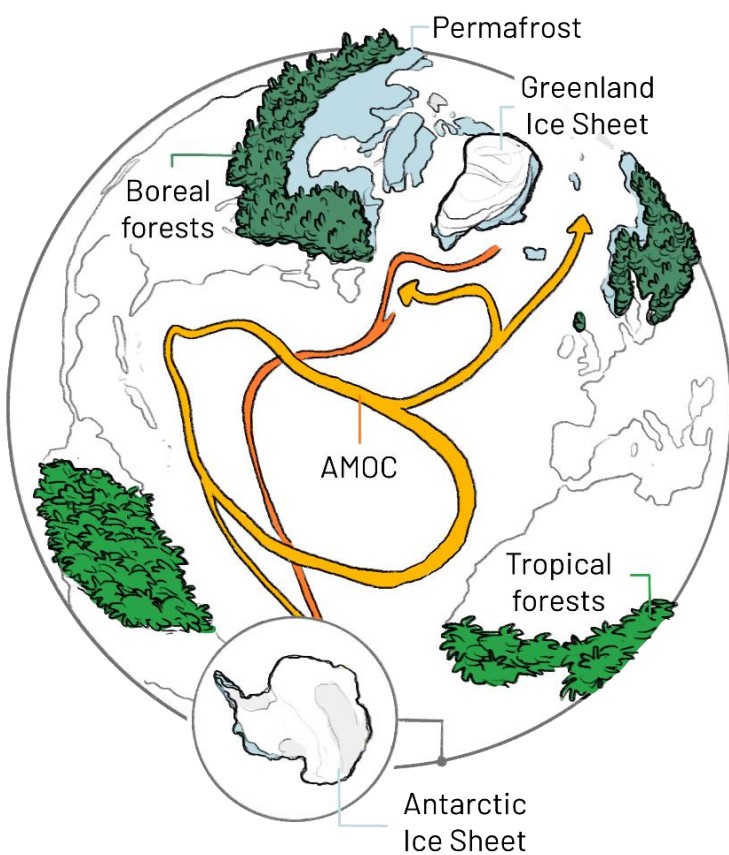

**Figure 2: Key tipping systems in TIPMIP.** In the first phase of TIPMIP, we will focus on potential tipping dynamics in the following systems: the Greenland and Antarctic ice sheets, permafrost, the Atlantic Meridional Overturning Circulation (AMOC), boreal forests, and tropical forests. We will further explore potential regional tipping dynamics, for instance in mountain glaciers, African semi-arid and arid regions (Sahel-Sahara), as well as terrestrial hydrological systems.

## 2 Tipping dynamics in the Earth system: definitions and characteristic traits

**Tipping dynamics** are triggered when a change in part of a system (**tipping system**) becomes self-perpetuating beyond some critical threshold (**tipping point**), leading to substantial, widespread, often abrupt and/or irreversible impacts (see Box 1). Examples for amplifying feedbacks which can trigger tipping dynamics in the Earth system include the ice-albedo feedback, permafrost-carbon feedback, melt-elevation feedback, moisture-recycling feedback or cloud feedback in a warming world (as further detailed in Section 3).



**BOX 1 – Terminology**
Definitions are based on IPCC-AR6, Brovkin et al. (2021), Armstrong McKay et al. (2022) and Lenton et al. (2023).

A **tipping system** is a large-scale component of the Earth system that can undergo an irreversible and/or abrupt shift when a **critical threshold (tipping point)** is crossed due to

gradual changes in external forcing or internal dynamics. This can lead to substantial, widespread and long-lasting physical, biophysical and/or biogeochemical impacts that in turn can induce far-reaching socioeconomic impacts.

**Tipping dynamics** comprise the dynamics associated with the transgression of a tipping point as well as the related feedback dynamics. A **tipping element** refers to a sub-continental to continental-scale tipping system.

**Forcing** refers to a driving factor that influences tipping system dynamics. For example, forcings considered in TIPMIP include incoming solar radiation, greenhouse gas emissions, aerosols and aerosol precursors, and land-use change. Forcing can either be direct human pressures (**Earth system-external forcing**, e.g., greenhouse gas emissions or deforestation) or changes within the Earth system (**Earth system-internal forcing**, e.g., temperature changes or internal variability such as decadal-scale changes in the ocean-atmosphere system). Note that Earth system-internal forcing might be external to a specific tipping system (e.g., AMOC variability might be an external forcing to the Amazon rainforest).

**Abrupt changes** in a system are large-scale changes that take place over a few decades or less or are occurring faster than the typical timescale of the system (e.g. decades for an abrupt AMOC collapse contrasted with multi-centuries to millennia for the circulation timescale of the AMOC), persist (or are anticipated to persist) for at least a few decades and cause substantial impacts in human and/or natural systems.

A **state variable** is a characteristic observable used to represent the overall state of a tipping system. For example, this could be the ice volume for the ice sheets or the overturning strength of the AMOC.

A system is in an **equilibrium state** when it has fully adjusted to any changes in forcing, such that changes in the state variables are negligible.

None of the Earth system components / potential tipping systems considered is in equilibrium at present and rapid changes in forcing prevent equilibrium states from being reached. Observed and future changes are thus a **transient response** to previous changes in forcing. In order to understand tipping dynamics in the Earth system we need to consider both the equilibrium as well as the transient responses.

Tipping can occur when forcing is increased beyond a critical level (for example as **bifurcation tipping**), or when forcing is increased too rapidly for the system to adjust, even though a critical (bifurcation) point has not been





passed (known as **rate-induced tipping**), or when variability (for example due to natural climate variability) causes a jump from one state to a substantially different one (known as **noise-induced tipping**).

Feedback loops refer to processes where a change in the system triggers effects that either amplify (**positive feedback**) or dampen (**negative feedback**) the initial change. Climate subsystems are subject to several feedbacks, both positive and negative, which interact in a complex manner. If positive feedbacks dominate, when approaching a critical forcing threshold, the amplification of initial changes can shift the system into another state.

**Committed/legacy impacts** are physical and biogeochemical impacts on the Earth system and the ecosystems in it, triggered in the coming years to decades, which then (due to slow response time scales and tipping dynamics) unfold over longer timescales, often centuries to millennia. For instance, once triggered, ice loss in Antarctica can continue for decades or even centuries, even if global mean temperatures are reduced back to values below a critical threshold.

**Hysteresis** describes a strong path-dependency of a system, where once a critical threshold is crossed, the forcing has to be reversed to significantly below this threshold before the system can recover or the respective state variable returns to its original value.

**Irreversibility** occurs when a system does not return to its original equilibrium state after the change in forcing is reversed to below the critical threshold.

**Early warning signals** are potential physical precursor phenomena or statistically derived quantitative indicators of approaching a tipping point, including changes in the spatio-temporal variability of observables such as ocean temperature, vegetation biomass, or ice sheet mass loss, commonly obtained by applying mathematical principles of dynamical systems to Earth system components. Early-warning can be measured in one-dimensional space using univariate precursors (for example, increasing temporal autocorrelation) or in multi-dimensional space (such as spatial patterns of vegetation cover) applying spatially explicit precursors.

Commonly, tipping systems are viewed as simple bistable systems, with two stable branches and one unstable branch in a
double fold bifurcation structure (see Fig. 3, left panel). This classical steady-state, one-dimensional understanding of tipping dynamics, however, may mask some of the more complex behaviours of Earth system components that are characterized by different kinds of spatial dynamics as well as different temporal dynamics. In the following, we give an overview of these potential complex behaviours which we expect to see in TIPMIP simulations, bringing together our understanding of Earth system dynamics with complex-systems analysis. With the planned TIPMIP experiments, we anticipate considerably
advancing not only our understanding of applied tipping dynamics in the Earth system, but also tipping theory more broadly.





In terms of the *temporal dynamics*, the classical steady-state view of tipping – triggered by very slowly forcing the system beyond a critical threshold in a quasi-equilibrium manner (**bifurcation tipping**), e.g. through increase in global warming beyond some critical level or due to internal variability in the system – needs to be complemented by other types of tipping mechanisms (Ashwin et al., 2012): in the Earth system components considered here, the forcing rates (e.g., rate of global warming) are typically much faster than the characteristic response timescales, which can lead to **rate-induced tipping** beyond a critical rate of change (e.g., Stocker and Schmittner, 1997; Wieczorek et al., 2023). Since changes in the Earth system are further strongly characterized by internal variability (Ghil and Lucarini, 2020), tipping can also be initiated through fluctuations close to a critical threshold (**noise-induced tipping**) (Ditlevsen and Johnsen, 2010) (see Fig. 3 right panel).

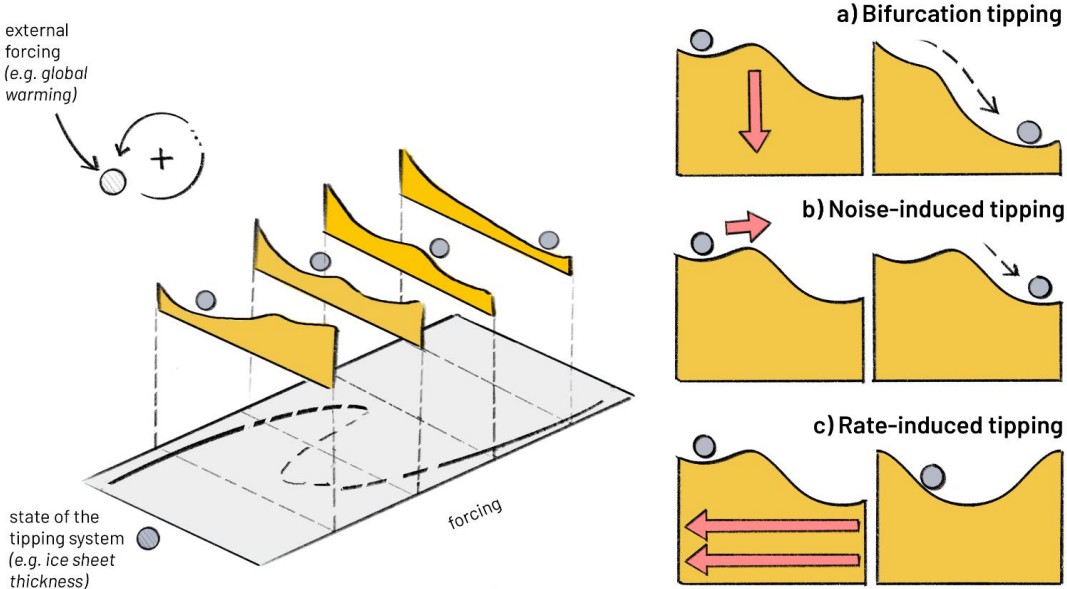

**Figure 3: Types of tipping.** Dynamical systems theory distinguishes at least three different types of tipping between co-existing, attracting states: **(a) Bifurcation tipping** is associated with a system bifurcation at some critical forcing level. Once that critical level is transgressed, the system will transition to an alternative (often less desirable) equilibrium state. **(b) Noise-induced tipping** occurs when stochastic fluctuations or internal variability are strong enough to push the system out of the basin of attraction of the original state. **(c) Rate-induced tipping** occurs when the external conditions vary faster than some critical rate of forcing change such that the system cannot follow fast enough; the transitions to the alternative state in this case can happen before crossing the critical (bifurcation tipping) forcing level. The right panel is based on van der Bolt and van Nes (2021).

In terms of the *spatial dynamics*, a more appropriate extension of the instructive, but rather conceptual perspective of a simple double fold bifurcation is to distinguish between tipping dynamics which play out as a fundamental change of an entire system (**macro tipping**), as tipping on a small scale spreading to large-scale changes (**propagating tipping**), and/or as independent





small-scale tipping synchronized on a larger scale **(clustered tipping)** (Lenton et al. 2024). These different spatial dynamics (see Fig. 4) are expected to also occur in the Earth system (e.g., Loriani et al., 2025).

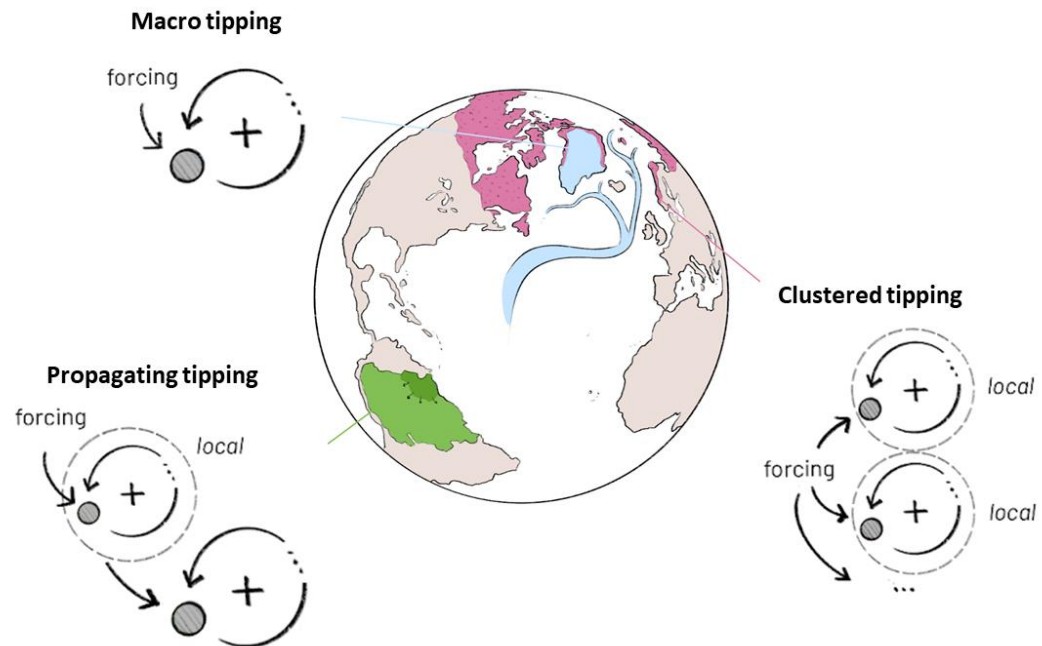

**Figure 4: Spatial properties of tipping dynamics in the Earth system. (a) Macro-tipping** constitutes the tipping of an entire system, typically driven by an amplifying feedback which causes large-scale change. An example of this could be the response of the Greenland Ice Sheet where, once a critical warming level is crossed, the melt-elevation feedback and other feedbacks can lead to large-scale decline of the ice sheet (e.g., Robinson et al., 2012; Bochow et al. 2023). **(b) Propagating**
**tipping** occurs when tipping at small scale spreads due to biophysical interactions. An example of this could be the Amazon rainforest, where the moisture-recycling feedback can lead to cascading changes in adjacent regions (e.g., Zemp et al., 2014; Staal et al., 2020b; Smith et al., 2023). **(c) Clustered tipping** describes multiple localized tipping events synchronized over large spatial scales. An example of this could be abrupt permafrost thaw occurring in multiple locations for the same global warming level (e.g., Turetsky et al., 2020; Nitzbon et al., 2024). Figure based on Loriani et al. (2025).

Approaching 1.5°C of global warming beyond pre-industrial levels, questions arise as to what this might entail in the long-term in terms of climate impacts, and especially their reversibility in context of different overshoot scenarios (Schleussner et al., 2024). Due to the lifetime of $CO_2$ in the atmosphere, climate change can be considered irreversible on human timescales (e.g., Solomon et al., 2009). More generally however, it is unclear to what extent certain Earth system components will adapt
or recover on which temporal and spatial scales - or, expressed in terms of the system dynamics - to what extent changes in specific state variables can be reversed with a reversal in forcing. Tipping dynamics might be triggered even when atmospheric $CO_2$ is reduced and global mean temperatures eventually sink again, depending on the magnitude and duration of the overshoot and the landing climate (Ritchie et al., 2019; Ritchie et al., 2021; Bochow et al., 2023; Wunderling et al., 2023).





**Figure 5: (Ir)reversibility of transient dynamics.** The diagram illustrates equilibrium (solid lines) and transient (dashed lines) responses to different forcing trajectories (here shown for the example of global mean temperature change). **(a) Overshoot scenario:** a transient forcing trajectory which follows a peak-and-decline scenario, eventually reaching the landing climate (the (quasi-)equilibrium warming level and corresponding changes in other climate variables). Panels (b)-(e) show possible transient (dashed blue curves) and committed (solid blue arrows) responses, assuming an overshoot scenario where the peak warming transgresses a critical threshold. In case the peak warming remains below the critical threshold, changes in the state variable are reversible, but the transient response can nonetheless diverge from the equilibrium response. **(b) Fully reversible:** a monostable system returns to its original state when a change in forcing is reversed and the system is given enough time to relax to equilibrium. **(c) Reversible:** In case of bistability, changes can still be reversible if the landing climate





is below the bistable range. **(d) Potentially irreversible:** If the landing climate is within the range of the bistable regime, the response depends on the duration and magnitude of peak warming. If the peak warming persists long enough, the trajectory will eventually approach the alternative state (upper branch), leading to irreversible impacts. For a shorter duration of peak warming, the trajectory can recover to its original state (lower branch). **(e) Irreversible:** If the landing climate is beyond the critical threshold, the system does not recover to its original state. Figure extended based on Klose & Winkelmann (in review).

This can also lead to a rich range of possible outcomes when crossing a critical threshold – from (fully) **reversible** to **potentially irreversible** to **irreversible** changes (see Fig. 5): If temperatures in a peak-and-decline trajectory are decreased but the long-term landing climate still exceeds a critical threshold, irreversible impacts occur. If the eventual landing climate is below the critical threshold, the impacts depend on the magnitude and duration of the peak warming (or warming above the critical threshold), where some trajectories still lead to irreversible impacts, while for other trajectories, substantial impacts might be avoided or reversed over long timescales. This range of trajectories has been shown to exist for instance for the Antarctic Ice Sheet (Klose & Winkelmann, in review).

Importantly, note that 'reversible' does not mean 'safe', i.e., significant impacts can still occur even if from a systems-dynamics viewpoint, the system recovers. Significant ice loss from Antarctica and consequent sea-level rise for instance can be caused even if the grounding line eventually recovers to its original position.

## 3 Key tipping systems and critical knowledge gaps

In the first phase of TIPMIP, we will focus on potential tipping dynamics in the following systems: the Greenland and Antarctic ice sheets, permafrost, the Atlantic Meridional Overturning Circulation (AMOC), boreal forests, and tropical forests. We will further explore potential regional tipping dynamics, for instance in mountain glaciers, African Semi-Arid and Arid Regions (Sahel-Sahara), and terrestrial hydrological systems. In a second phase of TIPMIP, we expect to add to this list.

This choice of focus systems is due to several reasons: (i) The underlying physics and feedback processes are relatively well-understood and included in state-of-the-art models. (ii) Transgressing critical thresholds even in parts of these core tipping elements would have far-reaching biophysical and socio-economic impacts (Armstrong McKay et al., 2022). (iii) Moreover, given the long response timescales, for instance of the ice sheets, some changes might be triggered in the coming decades, which then unfold over much longer timescales, leading to committed and/or irreversible impacts (Abrams et al., 2023; Golledge et al., 2015; Klose et al., 2024).

Current lines of evidence for the tipping systems considered here include paleoclimate records, observations and individual studies across the model complexity hierarchy (Dijkstra, 2024) based on advanced understanding of the underlying biophysical processes (Brovkin et al., 2021; Boers et al., 2022), as summarized below. TIPMIP will be the first systematic multi-model assessment of these tipping systems. In addition, we aim to study their interactions in coupled Earth system models, as the



interconnected nature of the Earth system could lead to potential cascading effects (Wunderling et al., 2024), but little is known about such tipping cascades on a process-detailed level.

### 3.1 Greenland and Antarctic Ice Sheets

The Greenland and Antarctic ice sheets are among the most prominent tipping systems: Evidence from paleo records (Blackburn et al. 2020; Turney et al. 2020; Christ et al. 2021) as well as ice-sheet models (Sutter et al. 2016; Robinson et al.,

2017) suggests that parts of both the Greenland and Antarctic ice sheets have undergone substantial change in the past, and there is an increasing risk of crossing tipping points with progressing anthropogenic climate change leading to self-sustained and potential irreversible ice loss (Winkelmann et al., 2023). Both the Greenland and Antarctic ice sheets are already losing mass, amounting to $169 \pm 16$ gigatons per year and $92\pm18$ gigatons per year between 1992 and 2020 (Otosaka et al., 2023), respectively, and are expected to become the largest sources of global sea-level rise in the future (e.g., Fox-Kemper et al.

2021).

Continental-scale Greenland Ice Sheet tipping is related to a feedback between decreasing surface mass balance and surface elevation of the ice sheet (e.g., Oerlemans, 1981). This melt-elevation feedback is expected to act on multi-centennial to millennial timescales at moderate warming levels (Robinson et al., 2012; Bochow et al. 2023). The Greenland Ice Sheet can therefore be considered an element with slow-onset tipping behaviour and its potential reversibility depends on duration and

225 magnitude of the warming.

Tipping dynamics of the Antarctic Ice Sheet are initially mostly linked to the potential instability of marine portions of the ice sheet (Reese et al., 2023): as ocean temperatures warm, basal melting below the floating ice shelves increases, leading to thinning. This may reduce buttressing of the inland ice and cause an increase of ice discharge into the ocean (Reese et al., 2017; Dupont and Alley, 2005; Fürst et al., 2016; Gudmundsson et al., 2019). Since many marine portions of the ice sheet are

230 subject to a retrograde bedrock slope, a positive feedback can be induced between thinning at the grounding line and increased ice loss (Marine Ice Sheet Instability; Weertman 1974; Schoof, 2007). As the Antarctic climate is considerably colder than the climate around Greenland, the Antarctic Ice Sheet is expected to enter a secondary tipping regime (related to surface mass balance processes similar to the Greenland Ice Sheet) only for substantial warming, after most of the ice in direct contact with the ocean would already be lost (Coulon et al., 2024; Klose et al., 2024).

The related feedback processes are comparably well understood, and their interplay gives rise to hysteresis behaviour in the ice-sheets' (quasi-)equilibrium response to warming on different spatial scales, as shown in previous modelling studies (Robinson et al. 2012; Garbe et al. 2020; Rosier et al., 2021; Reed et al., 2023, Höning et al., 2023). Major uncertainties, however, remain with respect to the critical temperature thresholds, the respective committed ice loss, its (ir)reversibility and related timescales: In the Amundsen Sea Basin of West Antarctica, recent studies suggest that the Marine Ice Sheet Instability

could either have been triggered already (Reese et al. 2023; van den Akker et al. 2025) or would be initiated with progressing warming by the end of the century, the bulk of potentially irreversible ice loss is expected to occur beyond the end of the



century (Golledge et al., 2015; Klose et al. 2024). So far, very few, individual modelling studies have covered such longer timescales (on the order of centuries to millennia), needed to capture tipping dynamics in ice sheets (Bochow et al. 2023; Golledge et al. 2015) and to complement projections of the ice-sheets' contribution to sea-level change on multi-decadal

timescales, as provided, for instance, by the Ice Sheet Model Intercomparison Project ISMIP6/7 (Nowicki et al. 2020; Goelzer et al. 2020; Seroussi et al. 2020).

Next to ice-sheet model uncertainties, substantial uncertainties are associated with the processes contributing to sustained and widespread ice loss, including surface runoff (Kittel et al., 2021; Fettweis et al., 2020) and firn (The Firn Symposium team et al., 2024), the sensitivity of sub-shelf melt to ocean warming (Coulon et al. 2024; Juarez-Martinez et al. 2024), as well as ice-

250 shelf collapse through damage (Sun and Gudmundsson, 2023) and hydrofracturing (Lai et al., 2020; Pollard et al., 2015; Scambos et al., 2009). In addition, limited observations give rise to uncertainties in, for example, basal friction and subglacial hydrology, that, among others, determine the sensitivity of the ice sheets to warming (Brondex et al., 2019; Sun et al., 2020; Dow, 2022; Kazmierczak et al., 2022). Recent developments of including dynamic ice sheets in Earth System Models (e.g., Mikolajewicz et al. 2007; Vizcaino et al. 2009; Gregory et al., 2020; Ackermann et al. 2020; Muntjewerf et al. 2021; Siahaan

et al., 2022; Delhasse et al., 2024; Goelzer et al., in discussion) to capture the two-way interaction with the climate system will eventually allow us to advance our understanding of tipping dynamics in ice sheets beyond the key feedback processes that have been identified so far.

In TIPMIP, we aim to assess these feedback processes, the potential for irreversible ice loss as well as the implications for long-term global sea-level rise in multi-model ensembles based on the experimental design outlined in Section 5.

## 3.2 Permafrost

Northern high-latitude permafrost (perennially frozen ground) is a major component of Earth's cryosphere, bearing various potentials for tipping dynamics due to its vast carbon and ground ice pools (Strauss et al., 2024). However, climate change is causing permafrost to thaw, exposing previously frozen carbon to increased microbial activity. This leads to enhanced

decomposition into $CO_2$ and $CH_4$, which can subsequently be released to the atmosphere. This positive feedback, the permafrost carbon feedback (Schuur et al., 2022), can further exacerbate global warming. However, its magnitude is still highly uncertain (Monteux et al., 2020; Schuur et al., 2022). In addition, another positive feedback between soil moisture, surface temperature, and cloud cover over permafrost regions in summer (permafrost-cloud feedback) could lead to substantial global warming (de Vrese et al., 2024). As such, both *abrupt* as well as widespread *gradual* (i.e., slow, top down) permafrost thaw

represents a critical Earth system risk exacerbating climate change impacts, including the release of vast quantities of greenhouse gases amplifying global warming and the reformation of entire ecosystems with associated risks for societies in the Arctic and beyond (Turetsky et al., 2020).



Limited paleo evidence suggests that permafrost regions were significantly affected by thaw during past periods of warming. These historical episodes underscore the potential for large-scale carbon release from thawing permafrost under warming

(Brovkin et al., 2021), though they do not provide clear thresholds or detailed feedback dynamics (Jones et al., 2023). That said, evidence for continental-scale tipping dynamics in permafrost remains sparse, though, localized abrupt thaw events are considered likely, due to regional feedback mechanisms, such as talik formation, thermokarst processes, or wildfires, which can drive rapid and sometimes irreversible thaw and thereby contribute to the destabilization of the climate system (Turetsky et al., 2020; Nitzbon et al., 2024). Current warming has already subjected parts of the permafrost region to thaw trajectories

(e.g., Farquharson et al., 2019; Nitze et al., 2020; Farquharson et al., 2022), and the likelihood of localized tipping grows with further global warming, especially beyond 1.5°C, suggesting that localized tipping thresholds could be crossed within decades if warming continues (McKay et al. 2022, Lenton et al., 2023).

Critical uncertainties arise from limited observational data and prevailing models' limited capability to simulate permafrost carbon release realistically. This is in part due to oversimplified or missing representation of physical and biogeochemical

processes such as abrupt permafrost thaw associated with thermokarst and thermal erosion, which drive and pace permafrost thaw at local and regional scales (Schädel et al., 2024). Specifically, models often omit subgrid-scale heterogeneities, lateral fluxes, dynamic landscape disturbances like thermokarst and wildfires, and excess ground-ice dynamics (Matthes et al., 2025).

On the other hand, even with improved process representations, there are still substantial uncertainties associated with the magnitude, distribution, and vulnerability of ground ice and organic carbon pools in the permafrost region. These are critical

for the preconditioning of permafrost landscapes to future thaw and carbon release, and specific tipping thresholds might arise from the imprint of climatic changes in the past (Nitzbon et al., 2020). Additionally, the combined influence of warming and other anthropogenic pressures such as deforestation on permafrost stability, as well as the interaction with other tipping elements, is not fully understood (e.g., Park et al., 2025). These process-based limitations hinder the models' ability to capture nonlinear feedbacks that are critical for understanding the role of permafrost in a warming world.

In TIPMIP, our objective is to evaluate the potential risks associated with abrupt and gradual permafrost thaw at both the regional and global scale. Since some of the non-linear mechanisms of abrupt thaw discussed above are as of yet not included in models and physical land-atmosphere feedbacks due to permafrost thaw are not well-studied with coupled models so far, the ability to assess the risk of large-scale abrupt thaw may be somewhat limited. However, it is clear that even in the potential absence of a critical threshold leading to large-scale abrupt thaw, the local and gradual impacts of permafrost thaw are

substantial, necessitating urgent mitigation and adaptation efforts (Nitzbon et al., 2024).

## 3.3 The Atlantic Meridional Overturning Circulation (AMOC)

The Atlantic Meridional Overturning Circulation (AMOC) plays a key role in the global heat and freshwater balance (Buckley and Marshall, 2016). Multiple lines of evidence from paleo records and numerical modelling across the model hierarchy



suggest that the AMOC may exhibit multistability (e.g., Willeit and Ganopolski, 2024), and could transition from a strong to a weak state due to the salt-advection feedback (Stommel 1961, Weijer et al., 2019; e.g. Hu et al., 2023; Hu et al., 2012; Rahmstorf et al., 1995; Ganopolski and Rahmstorf., 2001; Manabe and Stouffer, 1988) or convective instability (Rahmstorf, 1994; Wood et al., 1999; Kuhlbrodt et al., 2001; Swingedouw et al., 2021; Willeit et al., 2024) or internal variability (Romanou et al., 2023). Insights from past AMOC changes during glacial periods highlight the potential of rapid transitions (Rahmstorf,

2002) and abrupt change at regional (e.g., Heinrich events or D/O events) and global scale (e.g., ITCZ shifts, Timmermann et al., 2007; Orihuela-Pinto et al., 2022).

The stability and future evolution of the AMOC remains highly uncertain due to a combination of observational limitations, model biases, and incomplete understanding of key processes and feedbacks: Observational records, such as those from the RAPID array (Frajka-Williams et al., 2019), span only a few decades, making it difficult to distinguish long-term weakening

trends from internal variability (Bonnet et al., 2021; Latif et al., 2022). Climate models, while capturing broad features of the AMOC, show substantial divergence in the strength, structure, and sensitivity of the circulation, especially in response to global warming, freshwater inputs from the Greenland Ice Sheet, Arctic sea ice loss, and changing precipitation patterns (Jackson et al., 2022; Lee et al., 2021; Hu et al., 2011).

The magnitude of freshwater forcing required to trigger tipping dynamics of the AMOC is poorly constrained in both models

and data (Jackson et al., 2023; Hu, 2012, van Western et al., 2024). The processes that shape deep water formation, particularly the influence of overflows, convection, and mixing in the subpolar North Atlantic, are also represented with large uncertainties in Earth system models (Swingedouw et al., 2022). Furthermore, interactions between the AMOC, the atmosphere, and the cryosphere add complexity, with feedbacks between AMOC strength, storm tracks, sea-ice cover, and ice loss from the ice sheets playing a potentially amplifying role. For example, rapid atmospheric warming alone can substantially weaken the

AMOC, without the need for changes in surface freshwater flux (Gregory et al., 2005; Weaver et al., 2007; Levang and Schmitt, 2020).

In TIPMIP, building on other model intercomparison projects such as NAHosMIP (Jackson, 2023), we aim to investigate the combined effects of global warming and freshwater forcing on the AMOC, and assess the risk of AMOC weakening as well as subsequent impacts.

### 3.4 Tropical forests

Tropical forests cover approximately 1.95 billion hectares globally, including both mature and degraded areas, and represent crucial components of the Earth system (Pan et al., 2011). They harbour a disproportionately large share of the planet's biodiversity (Slik et al., 2015; Pillay et al., 2021) and are home to many Indigenous peoples and local communities, reflecting

high biocultural diversity and a long history of human habitation (Ellis et al., 2021). Tropical forests, such as the Amazon forest, also play a direct critical role in the Earth system, not least due to their potential to regulate global and regional climate



change through carbon storage, evapotranspiration, their influence on atmospheric circulation patterns such as the Intertropical Convergence Zone, and their role in providing ecosystem services, as hotspots of biodiversity and their impact on atmospheric composition (SPA 2021, chapter 7; Beveridge et al., 2024). At present, tropical forests store 471±93 GtC of carbon including
340 soil carbon (Friedlingstein et al., 2025). Anthropogenic climate change and land-use change threaten the tropical forest cover, and have already led to substantial changes in terms of biosphere integrity (Costa et al., 2021; Lapola et al., 2023; Flores et al., 2024). In particular, parts of the Amazon forest in the southeast region have degraded and may even have turned from a carbon sink to a carbon source in response to more intense dry seasons, temperature increases and deforestation (Gatti et al. 2021; Gatti et al. 2023).

Tropical forests have been identified as potential tipping systems, where the different forest portions could transition to different degraded ecosystems depending on the disturbances at play (Flores et al., 2024). This is due to multiple amplifying feedbacks, including the moisture-recycling feedback: if forest cover decreases (due to climate change, deforestation, fire or pests), the evapotranspiration decreases as well, leading to reduced regional rainfall which can affect adjacent regions (Staal et al., 2020b; Zemp et al., 2014; Smith et al., 2023). Another key feedback mechanism is the fire feedback: deforestation,
fragmentation, and climate change increase the likelihood of more frequent and intense fires which in turn reduces canopy cover, increases forest floor dryness, and promotes the spread of flammable invasive grasses, further enhancing the likelihood of future fires (Drüke et al., 2023; Cochrane et al., 1999; Martinez-Cano et al., 2022, Staver et al., 2011). Additionally, the loss of biodiversity and ecological interactions — such as seed dispersal by animals and species that confer drought resilience — can erode ecological buffering capacity, making forests more vulnerable to crossing thresholds (Bai and Tang, 2024).

Despite growing recognition of these processes, significant uncertainties remain in predicting if, when, and where tropical forest tipping dynamics may occur. In particular, the response to combined climate and land-use pressures is poorly constrained. Key structural uncertainties arise from limited abilities to simulate soil water dynamics and limited availability of observational data (Xu et al., 2013; Oberpriller et al., 2021), the neglect of  processes in model simulations, such as forest fragmentation and edge effects which could accelerate drying and fire spread (Bauer et al., 2024), the uncertain strength of the
$CO_2$ fertilization effect (Norby et al., 2010; de Almeide et al., 2016; Fleischer and Terrer, 2022), assumptions of tree susceptibility to drought and extreme events, generalized non-locally adapted representation of vegetation by the use of plant functional types (PFTs), assumptions on strength and velocity of adaptive capacities of forests e.g. by compositional adjustments, and missing secondary effects of mortality like pest outbreaks (Kolus et al., 2019; Sakschewski et al., 2021; McDowell et al., 2022) .

Earth system models typically do not resolve most of those relevant vegetation dynamics. Some more sophisticated vegetation models might resolve them but are mostly standalone models and hence do not account for important feedback mechanisms between vegetation and other parts of the Earth system such as the atmosphere. Therefore, a systematic understanding of the temporal and spatial dynamics with respect to the recovery of forests and hysteresis behaviour remains unclear (Staal et al., 2020a).





In TIPMIP, we will assess the thresholds of tropical forests for large-scale tipping dynamics by systematically testing different drought, heat, and $CO_2$ scenarios as well as different levels of drought and heat susceptibility, disturbance (e.g. through fires or deforestation), $CO_2$-fertilization strength and adaptive capacities of forest communities in dynamic global vegetation models. Lessons learned will then inform future experimental set-ups of ESMs participating in TIPMIP to include dynamic (rather than prescribed) vegetation changes (as the current vegetation components in ESMs are expected to be too stable).

### 3.5 Boreal forests

Boreal forests, covering extensive areas of the northern hemisphere, store approximately $272 \pm 23$ GtC, primarily below ground, making them crucial to global climate regulation (Pan et al., 2011), among others via albedo changes (Lawrence et al., 2022). Climate change poses significant threats to these ecosystems, potentially triggering tipping points leading to large-scale

biome shifts (Armstrong McKay et al., 2022). Two major scenarios have been identified: the dieback of southern boreal forests and the northward expansion into tundra regions.

Southern boreal forests face increased frequency and severity of disturbances such as wildfires, insect outbreaks, droughts, and logging, challenging their resilience (Rotbarth et al., 2024). For instance, intensified fire regimes in Siberia and North America have already led to forest regeneration failures (Burrell et al., 2022; Whitman et al., 2019). Tree mortality is amplified

by compounded disturbances, such as drought weakening trees against pests (Frelich et al., 2021; Stevens-Rumann et al., 2022). Climate models suggest fire frequency could increase by up to 50% in boreal Canada by the late 21st century, favouring broadleaved over coniferous species, potentially transforming these forests into open woodlands or grasslands (Price et al., 2013; Anoszko et al., 2022).

Conversely, warming may facilitate a northward migration of boreal forests into tundra regions (Rotbarth et al., 2024), as also

indicated by paleo data and modelling (CAPE, 2006; Sommers et al., 2021). While satellite observations indicate increased tree density within existing boreal regions, evidence for significant northward migration remains limited, failing to compensate for southern losses (Rotbarth et al., 2024). Additionally, shrubification in tundra ecosystems and positive feedback between shrubs and tree seedlings could enhance northern expansion (Mekkonen et al., 2021).

Significant uncertainties persist, notably regarding the magnitude of $CO_2$ fertilization effects, biodiversity impacts on

resilience, and inadequately modelled tree mortality from disturbances (Reich et al., 2022). Enhanced observational studies and improved model representations of mortality and regeneration processes are crucial to reduce these uncertainties (Seidl et al., 2017).

In summary, boreal forests are approaching critical climate-induced tipping points, with southern dieback scenarios being increasingly probable under moderate warming. Immediate action in climate mitigation and improved forest management

practices are essential to prevent large-scale transformations and associated carbon release.





In TIPMIP, we will assess the thresholds for large-scale boreal forest dieback and northern migration into tundra by systematically testing different warming scenarios, disturbance regimes (fire frequency, insect outbreaks), moisture availability (drought intensity and duration), levels of $CO_2$ fertilization strength, permafrost interaction effects, regeneration capacities, and species compositional shifts (coniferous vs. broadleaf dominance) in dynamic global vegetation models. Results will also
guide subsequent experimental configurations of Earth System Models (ESMs) participating in TIPMIP and other MIPs.

## 3.6 Regional tipping systems

In addition to the core tipping systems discussed above, we will further assess risks associated with regional tipping dynamics - in particular, we aim to quantify the potential for irreversible impacts from mountain glaciers, the African semi-arid and arid
regions of northern Africa (Sahel-Sahara) as well as terrestrial hydrological systems.

Studies on the potential tipping behaviour of **non-polar glaciers** suggest that, overall, glaciers outside the Greenland and Antarctic ice sheets are expected to respond linearly to temperature changes throughout this century (Rounce et al., 2023). However, at higher warming rates (Bolibar et al., 2022), over longer timescales (Marzeion et al., 2018), and at the scale of individual glaciers, ice caps, and icefields (Åkesson et al., 2017; Davies et al., 2022, 2024; Zekollari et al., 2017), these smaller
ice bodies may exhibit more nonlinear dynamic responses to climate-induced warming. Due to these complexities, the tipping potential, timing, and dynamics of mountain glaciers remain less well understood and are likely driven by a combination of locally-specific factors (Kääb et al., 2023). In TIPMIP we aim to assess the tipping potential and respective biophysical impacts at a regional scale.

The West African Monsoon (WAM) governs hydroclimatic variability, vegetation dynamics, and dust emissions across
northern tropical Africa, including the **Sahel-Sahara region**. Paleoclimate records reveal dramatic shifts in WAM intensity, such as the African Humid Periods (AHPs), when vegetation expanded into the Sahara, and abrupt droughts linked to interactions between orbital forcing, AMOC variability, and regional feedbacks like albedo-vegetation-precipitation loops (deMenocal et al., 2000; Shanahan et al., 2015; Pausata et al., 2020; Dallmeyer et al., 2021). Modern anthropogenic warming introduces new risks: land-use changes (e.g., desertification) amplify albedo-driven radiative feedbacks, suppressing
convective rainfall (Charney et al., 1975; Zeng et al., 1999), while AMOC weakening could destabilize the Intertropical Convergence Zone (ITCZ), driving abrupt monsoon shifts (Gupta et al., 2003; Masson-Delmotte et al., 2021). Large-scale geoengineering initiatives like the Great Green Wall (GGW) – aimed at restoring Sahelian ecosystems – share some characteristics of past greening mechanisms (e.g., vegetation-albedo feedbacks) and may mitigate desertification (Pausata et al., 2020). However, recent modelling shows such efforts risk unintended trade-offs, including reduced drought duration
alongside intensified pre-monsoonal heat extremes (Ingrosso and Pausata, 2024). Although model projections disagree on tipping thresholds (estimated ~2°C warming) and timescales (decadal to centennial), paleoclimate evidence and amplifying feedbacks (soil moisture loss, dust-climate interactions) classify WAM as a potential tipping system with low confidence





(Armstrong McKay et al., 2022). Critical knowledge gaps persist in resolving vegetation-climate feedbacks, aerosol impacts, and regional heterogeneity in abrupt responses, necessitating targeted research to constrain risks of irreversible Sahel-Sahara

hydrological transitions. These gaps will be addressed in targeted AOGCM and ESM experiments in TIPMIP.

Tipping dynamics in the **terrestrial hydrological system** are particularly evident in hydrological extremes, such as transitions from manageable water scarcity to severe drought, or from stable river flow regimes to catastrophic floods. Tipping points in terrestrial water systems arise from the complex interplay of hydrological, climatic, ecological, and anthropogenic factors, often amplified by feedback loops (e.g., wet-dry swing: Madakumbura et al., 2019; frontal precipitation: Moon et al., 2023;

aridity: Takeshima et al., 2020; tropical cyclones: Utsumi and Kim, 2022; land-atmosphere interaction: Zhang et al., 2020). For instance, prolonged drought conditions can reduce vegetation cover, decreasing soil water retention and further intensifying aridity - a process known as desertification tipping (Scheffer et al., 2001). Similarly, glacier retreat in response to warming can abruptly alter river flows, impacting downstream water availability (Pritchard, 2019).

In TIPMIP, critical thresholds in terrestrial hydrological systems will be assessed, where small changes in climate, land use,

or human activity can lead to significant and often irreversible shifts in water availability, quality, and distribution. Furthermore, by integrating climate science, hydrology, disaster risk management, and socio-economic analysis, this domain seeks to identify early warning signals of tipping points, assess their potential ecological and socio-economic impacts, and develop strategies for mitigation and adaptation.

## 4 Overarching aims and key research questions for TIPMIP

The overarching aim of TIPMIP is to improve scientific understanding of nonlinear changes in the Earth system, quantify respective uncertainties and assess the characteristic traits and risks associated with tipping dynamics in the Earth system (see definitions in Box 1). This requires quantifying the likelihood of transgressing critical thresholds or tipping points in individual Earth system components, as well as the respective biophysical and biogeochemical impacts on different timescales. Based on this analysis of likelihood and impacts within TIPMIP, we aim to generate an exposure risk assessment, i.e., an assessment of

how many people could be exposed to biophysical impacts (such as sea-level rise, reduction in net primary production etc.) due to tipping dynamics in the future. Our results can then form the basis for studying the respective socio-economic impacts (e.g., agricultural losses, damages to infrastructures and settlements, reductions in economic growth) in collaboration with the impacts modelling community for instance in ISIMIP (Frieler et al., 2017), as indicated in the outlook (see Fig. 8). Acknowledging the differences in vulnerability in different world regions is important to gain a comprehensive picture of

response options and adaptive capacities to tipping risks. Our community aims to address such questions together with social scientists in the future, but this research is beyond the scope of TIPMIP itself.

As tipping dynamics are associated with traits of abruptness, critical threshold behaviour, multistability and hysteresis, the experiments within TIPMIP are specifically designed to further our understanding of these characteristics.





## BOX 2 – Key research questions in TIPMIP

**Key questions** to be addressed in TIPMIP are:

- Which are the **key amplifying and stabilizing feedbacks** associated with tipping dynamics, and their respective feedback strengths?

- Are there **critical thresholds, beyond which tipping dynamics are triggered**? If so, what is the likelihood of crossing these critical thresholds at different levels of global warming and/or land-use change?

- Once critical thresholds are transgressed, what are the resulting **immediate and committed biophysical impacts over different timescales** in different parts of the world?

- What role does the **forcing rate** and **forcing duration** play for tipping dynamics?

- Under which conditions and to which extent are these biophysical impacts **(ir)reversible**, and on which timescales?

- What role do the complex **interactions** between tipping systems play?

- What are the associated **uncertainties** and how can they be quantified?

- What are relevant **structural and process limitations** of (coupled) models with respect to their ability to represent the dynamics of tipping elements?

- What are the prospects for operational **early warning indicators** of approaching critical thresholds that could inform action for adaptation or resilience?

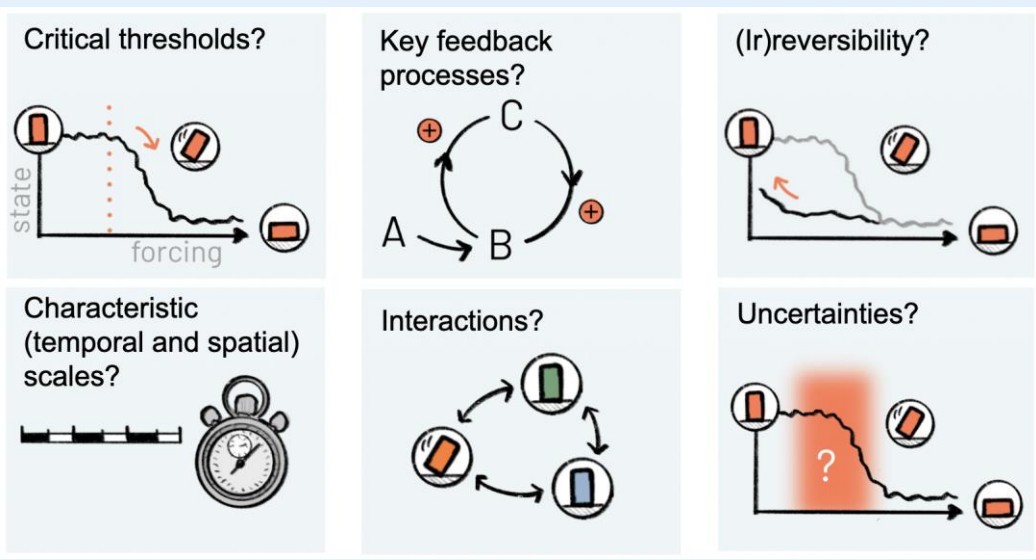





TIPMIP is the first systematic initiative to address these key questions in a standardized **multi-model assessment across**
**multiple components of the Earth system in atmosphere, oceans, cryosphere and biosphere**, including coupled Earth
System Models (ESMs) as well as stand-alone models such as Ocean Models (OMs), Ice-Sheet Models (ISMs), Land Surface
Models (LSMs) and Dynamic Global Vegetation Models (DGVMs).

We are aiming at addressing three different timescales within our TIPMIP experiments (Fig. 6), looking at changes over the
next decades (**policy timescale**), committed impacts which could occur over the next millennium (**commitment timescale**),
and long-term impacts which impede the functioning of critical Earth system components (**stability timescale**).

These goals are ambitious, as – due to the very nature of tipping dynamics, associated with highly non-linear responses of parts
of the Earth system – the risks are very difficult to quantify. Major uncertainties arise due to:

- an incomplete understanding and quantification of the underlying feedbacks and/or interaction between different
processes in the Earth system,
- some interactions and feedbacks not being represented in models - e.g., ESMs not including interactively-coupled ice
sheets or a fully dynamic carbon cycle,
- some parametrizations not capturing the full range of dynamics in Earth system components,
- models potentially being overly stable to capture tipping dynamics (Valdes, 2011; Liu et al., 2017; Fox-Kemper et
al., 2021),
- the (spatial and/or temporal) model resolution not being sufficient to capture certain processes and/or biophysical
impacts,
- lack of adequately resolved observational data for model input and validation, especially on longer timescales,
- the computational costs of the envisioned simulations limiting the ensemble size and number of simulation years
(both important to address tipping risks).

Despite these uncertainties, we expect that a wealth of information will be gained from the TIPMIP experiments, yielding the
first systematic assessment of tipping dynamics across different models and across different key components of the Earth
system, based on the same experimental design. TIPMIP will further help us identify key knowledge gaps which can prompt
targeted model development in the future.




# 5 TIPMIP experimental design and structure

## 5.1 Experiment Types

TIPMIP's key research questions will be approached via six main types of simulation experiments performed using coupled
Earth system models as well as stand-alone domain models (see Fig. 6 and Table 1). In a first set of experiments, we increase
the climate forcing ($CO_2$ emissions or concentrations, depending on the type of model) starting from pre-industrial conditions
(**climate ramp-up / warming experiments, E1**). To this end, ESMs are forced using a $CO_2$ emission rate that (based on the
respective transient climate response to cumulative emissions of carbon dioxide (TCRE) for each individual model) gives a
linear increase in global mean surface air temperature of about 0.2 degrees per decade. We then branch off at different levels
of global warming to study the respective likelihood of triggering tipping dynamics and their long-term, committed impacts
(**climate stabilization / commitment experiments, E2**). The reversibility of these changes and related timescales are assessed
by then reducing the climate forcing to various landing climates (**climate reversibility / cooling experiments, E3**). We further
propose to assess the forcing-rate dependency of tipping dynamics by varying the rate during the climate ramp-up / warming
and reversibility / cooling experiments (**climate rate experiments, E4**).

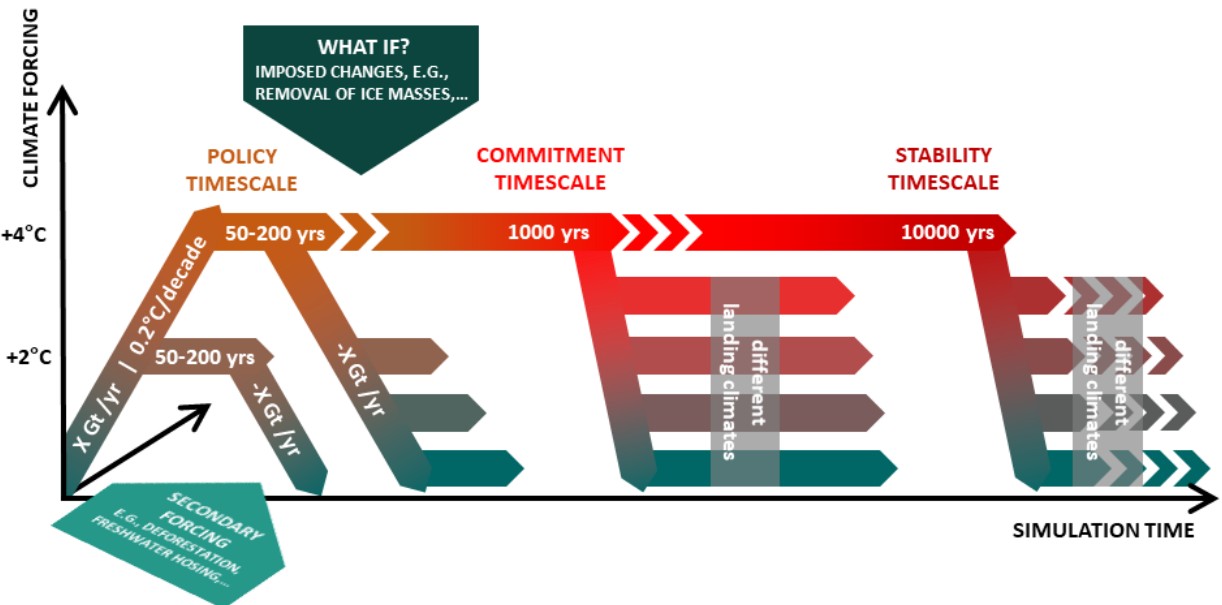

**Figure 6: TIPMIP experiment types.** See Table 1 for further description.

In addition to these four **climate forcing experiments (E1-E4)**, we will perform **secondary forcing experiments (E5)** by
applying additional system-specific forcing (for instance, freshwater hosing in the North Atlantic or deforestation in the
Amazon rainforest). **What-if experiments (E6)** will explore the effects of a complete state-change of a tipping system on the
510 dynamics of the whole Earth system (for instance, by removing the Greenland ice cover or Amazon rainforest).



**Table 1**: **Description of TIPMIP experiments as well as the research questions and Earth system components (TIPMIP domains) they address.**

| Experiment type | Description | Research questions addressed | TIPMIP Domain in which this experiment is run |
|---|---|---|---|
| **Climate ramp-up / warming (E1)** | Ramp up of climate forcing; ESMs use a $CO_2$ emission rate that, based on the model TCRE, gives a linear increase in global mean surface air temperature of ~0.2K/decade. | Baseline | all |
| **Climate stabilization / commitment (E2)** | Branch off the climate ramp-up / warming experiment (E1) simulations at different levels of global warming relative to the pre-industrial, and extend with zero $CO_2$ emissions. | Likelihood of crossing critical thresholds? Immediate and committed biophysical impacts? | all (extended to different timescales depending on domain and computational feasibility) |
| **Climate reversibility / cooling (E3)** | Branch off the climate stabilization experiment (E2) simulations and apply negative $CO_2$ emissions of the same magnitude as used in the climate ramp-up experiments. Simulations can be further extended at different landing climates (global mean temperature levels). | (Ir-)reversibility and hysteresis of tipping dynamics and impacts? | all |
| **Climate rate-dependency (E4)** | Ramp up of global mean temperature at slower or faster rate compared to E1. | Forcing rate-dependency of tipping dynamics? | all |
| **Secondary forcing (E5)** | Additional non-GHG forcing, e.g., land-use change or freshwater hosing. | Resilience of tipping systems under multiple forcings? | TIPMIP-OCEAN, TIPMIP-BIOSPHERE, TIPMIP-PERMAFROST, TIPMIP-REGIONAL |
| **What-if (E6)** | Imposing a complete state-change of part of the Earth system, e.g., removing Greenland ice cover or Amazon rainforest cover. | If an Earth system component tips, what are the impacts on the rest of the Earth system? | all |



In a unified approach, we will apply this same set of experiments to all tipping systems considered in TIPMIP. Since TIPMIP will encompass coupled ESM simulations (both in emissions and concentration driven mode), as well as stand-alone model simulations, the detailed application of these joint experiments will be outlined in individual protocol papers, along with the input variables, CMORization standards (Mauzey et al. 2024) where applicable, participating models and model prerequisites,

as well as the metrics for model output analysis. An initial TIPMIP-ESM Tier 1 experimental protocol is described in Jones et al. (in prep.), the other domain protocols are underway.

## 5.2 TIPMIP structure and community

TIPMIP is an international, community-driven effort based on the initiative and expertise of its many participants. TIPMIP is closely connected to the World Climate Research Programs (WCRP)-Coupled Model Intercomparison Project (CMIP) and is

a CMIP7-registered MIP, CMIP7 constituting the next phase of the Coupled Model Intercomparison Project that provides the quantitative basis of climate model simulations for the upcoming IPCC AR7 report. Results and outputs from TIPMIP will be provided by the individual modelling groups and the outputs will be standardized in accordance with accepted CMIP protocols (CMOR version 3.9.0, Mauzey et al. 2024). This will allow all TIPMIP outputs to be provided as open access data sets on the Earth System Grid Federation (ESGF). Model signup can be achieved either via the TIPMIP website (www.tipmip.org), or by

contacting the TIPMIP core team and domain leads.

In order to facilitate the design, conduction and analysis of simulations, TIPMIP is organized in several **domains** (see Fig. 7).

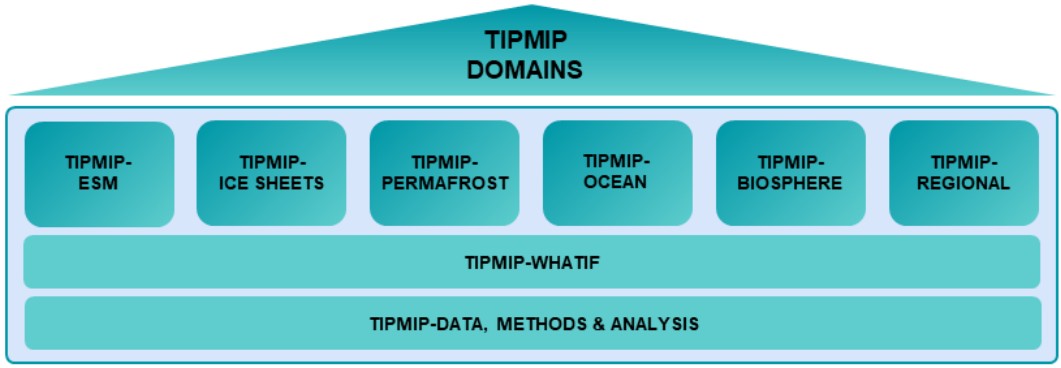

**Figure 7: Organizational structure of TIPMIP.** As TIPMIP is essentially a MIP of MIPs, focusing on different tipping systems with coupled ESMs and stand-alone models, the community is structured into according domains. A brief description

of each domain is given below; a more detailed workflow and progress for each domain is further described on the TIPMIP webpage.



**TIPMIP-ESM** performs and analyses the experiments with fully coupled Earth System Models (see Table 1), both in $CO_2$-emission- and $CO_2$-concentration mode. The Climate Forcing Tier 1 experiments are described in detail in Jones et al. (in prep.). Due to the computational constraints, typical simulation timescales are on the order of centuries. In a Tier 2 set of experiments, extensions to these are planned to bring in more realistic forcing (e.g. aerosols, land-use), different landing climates off the negative emission runs (to look at reversibility) and to increase the ensemble size. To explore variability in larger ensembles and across longer timescales, analogous simulations will be performed with Earth System Models of Intermediate Complexity (EMICs).

**TIPMIP-ICE SHEETS** performs and analyses experiments with stand-alone Ice Sheet Models. These experiments will use TIPMIP-ESM output to force the Ice Sheet Models with respect to surface and ocean temperatures and surface mass balance changes. Focus will be on climate-forcing experiments (E1-E4), extending the typical timescales of the TIPMIP-ESM experiments to multiple millennia to account for the longer response times of ice sheets (covering policy, commitment and stability timescales).

**TIPMIP-PERMAFROST** performs and analyses experiments with ESMs and stand-alone (uncoupled) land surface models (in which some permafrost processes are better represented, but where only experiments in $CO_2$ concentration mode is possible) which can provide permafrost-relevant diagnostics such as soil temperature, soil water/ice contents and soil organic carbon contents. One focus will be on identifying macro-tipping thresholds in the Climate Forcing experiments. In addition, the possibility for local, regional, or clustered tipping will be explored in Secondary Forcing experiments using the stand-alone models.

**TIPMIP-OCEAN** performs and analyses experiments with ESMs and AOGCMs. Studies of AMOC tipping have shown a great sensitivity to the amount of freshwater added (e.g. through ice loss from Greenland), so the focus of this domain includes Secondary Forcing experiments to explore this sensitivity to freshwater input. This includes both idealized freshwater hosing experiments and more realistic experiments based on past and projected Greenland ice loss.

**TIPMIP-BIOSPHERE** performs and analyses experiments with dynamic global vegetation models. The focus will be on Climate Forcing as well as Secondary Forcing experiments. These secondary forcings include deforestation, drought, fires, nutrient limitations and increased mortality. A further aim of the experiments is to quantify the key uncertainty posed by the unknown potential of the $CO_2$ fertilization effect.

**TIPMIP-REGIONAL** encompasses experiments focusing on other tipping systems, such as the mountain glaciers or terrestrial hydrological systems. For the latter, experiments will be performed with stand-alone land surface models and global hydrologic models with a focus on diagnosing and analysing tipping dynamics related to terrestrial water resources and water-related disasters, such as floods and droughts, as a complementary extension of the TIPMIP-ESM experiments.



**TIPMIP-WHATIF** designs and analyses experiments with AOGCMs and ESMs to quantify cascading climate risks and extreme storylines arising from the future tipping of critical Earth system components. These experiments will impose changes in commitment simulations, such as removing Greenland ice cover, Amazon rainforest, or by exploring climate responses to drastic permafrost thaw.

**TIPMIP-METHODS** develops approaches to automatically detect spatio-temporally coherent clusters of nonlinear / abrupt system changes, corresponding potential early warning signals at various scales, as well as other methods such as emulator models for systematic risk analysis based on TIPMIP outputs.

## 6 Analysis of TIPMIP simulations

The results of the TIPMIP ensemble experiments will allow to identify and evaluate the participating ESM's and domain models' representation of key Earth system processes that are known to be important for (either) stabilizing or destabilizing the potential tipping phenomena we are interested in, as well as evaluating the simulated underpinning processes of the (potential tipping) systems themselves (Jones et al., in prep.). More specifically, TIPMIP will help to (i) identify if/why ESMs and domain models are overly (un)stable, (ii) understand if and which simulated tipping dynamics are physically, biogeochemically or ecologically plausible, (iii) identify which modelling approaches are best-suited to represent tipping dynamics.

Furthermore, the occurrence of nonlinear responses to forcing can go unnoticed in complex models which generate a large amount of data in the form of fields of time series anchored to thousands of spatial grid cells and dozens of variables (Drijfhout et al., 2015). Hence, the large variety of models, experiments, and diagnostic output variables that are part of TIPMIP calls for standardized methods that can systematically screen the datasets for signals associated with tipping dynamics. The tell-tale signature of such nonlinear transitions in a system exposed to external forcing is an abrupt shift in the equilibrium response that can conceivably be attributed to a major positive feedback (see Box 1). In principle, the commitment experiments (E2) provide this equilibrium response to different forcing levels, however they yield a limited set of data points only and are comparatively expensive simulations. For sufficiently slow changes in forcing, however, the transient response of the system can provide insights into potential tipping dynamics, too. This means that some evidence regarding tipping points can be gathered by looking for abrupt shifts with respect to time (rather than forcing) in the outputs of the transient ramp experiments (E1).

The fact that abrupt shifts in a system can follow a trajectory with a very similar shape regardless of the absolute time scale or underlying mechanism, allows one to design automatic data mining methods which can automatically extract abrupt shifts from complex and large data. These events could be very difficult to find otherwise, in particular regarding shifts in unexpected regions or variables, where one does not know a priori where to look. In principle, such methods have a long tradition in and





far beyond climate sciences (see Bathiany et al., 2024), for example in the context of quality control and detecting breakpoints.

Change-point detection methods (Basseville and Nikiforov, 1993; Beaulieu et al., 2012), Bayesian change-point detection (Perreault et al., 2000; Ducré-Robitaille et al., 2003), or simple gradient-above-threshold approaches (Drijfhout et al., 2015) have been traditionally applied to single (univariate) time series. The aforementioned approaches commonly focus on abrupt changes in mean, variance, or linear trend behaviours. However, bifurcations in dynamical systems and, hence, corresponding tipping dynamics may also manifest exclusively in higher-order dynamical characteristics beyond mean, variance, or

autocorrelation structure undergoing sudden shifts. There exist a plethora of concepts from nonlinear time series analysis that characterize such higher-order features and could potentially allow to detect associated more subtle types of regime shifts in Earth's subsystems. One prominent example for such complex systems-based techniques is recurrence analysis that is able to detect nonlinear shifts in higher-order time series properties such as determinism, laminarity or effective dimensionality of the underlying system dynamics (Marwan et al., 2009; Donges et al., 2011).

Once individual tipping events have been identified in the simulations at grid point level, spatial clusters of tipping events can be ultimately identified using spatio-temporally aware methods such as edge detection (Canny, 1986; Bathiany et al., 2020), which are subject to active research (Terpstra et al., 2024; Loriani et al., 2024). The identified clusters can subsequently serve as analysis tools for other runs, e.g. as masks for the evaluation of the hysteresis experiments (E3).

While such methods serve as an indispensable introspection tool into the vast datasets expected to be produced in TIPMIP, the

610 final evaluation with respect to tipping points will be conducted in close collaboration with the respective domain and model experts. The identified clusters of interest need to be scrutinised against conceivable positive feedback loops acting in those regions and system domains and against model characteristics such as hard-coded thresholds that can yield an abrupt shift wrongly associated with tipping dynamics.

Besides the actual (retrospective) detection of tipping behaviour in the generated TIPMIP model output, the targeted

simulations obtained for the different potential tipping systems will also allow further exploring the potential of early warning indicators in identifying precursor signatures of especially incipient bifurcation tipping. Such indicators may include classical univariate measures like variance and temporal autocorrelation (i.e., critical slowing down, cf. Held and Kleinen, 2004; Dakos et al., 2008), but also spatiotemporal properties such as spatial autocorrelation or percolation processes in correlation based network representations of spatiotemporal fields of characteristic variables, such as ocean temperature fluctuations in case of

abrupt changes in the AMOC (van der Mheen et al., 2013). While such elaboration on early warning signals is not a core part of TIPMIP, the generated wealth of data will be available for interested members of the scientific community for corresponding follow-up studies.



## 7 Outlook and getting involved

In this overarching description paper, we give an overview of the current theory, understanding and knowledge gaps with respect to tipping dynamics in the Earth system; and present the plans and experimental design for a **first global atlas of potential tipping dynamics in the Earth system** via the multi-facetted, multi-model assessment TIPMIP. The experiments are geared to answer pressing questions, for instance on the long-term committed impacts of greenhouse gas emissions and anthropogenic climate change, on the (ir)reversibility of impacts, or on the role of warming rates (or the rate of change of other 630 forcings). The TIPMIP-WHATIF experiments will further shed light on the consequences of tipping point transgressions by imposing changes in one part of the Earth system and analysing the response of other parts in coupled experiments.

Although we initially focus on the Greenland and Antarctic ice sheets, permafrost, the Atlantic Meridional Overturning Circulation (AMOC), boreal forests, and tropical forests as well as potential regional tipping dynamics (for example in mountain glaciers, semi-arid and arid regions, and terrestrial hydrological systems), our approach could be extended to also 635 cover non-linear change and committed impacts in other Earth system components in the future (for an overview see Armstrong-McKay et al., 2022; Lenton et al., 2023).

TIPMIP is CMIP-registered and embedded in the international research landscape, drawing on state-of-the-art Earth system as well as domain-specific models. Given the wide scientific and public interest and debate on tipping dynamics in the Earth system on its unfolding Anthropocene trajectory (e.g., Steffen et al., 2018), it is important to investigate these in a solidified 640 manner. We have therefore chosen to start our analysis with idealized experiments, which will allow us to disentangle the characteristic traits of the very different tipping systems considered. More realistic scenarios are planned in a second phase of TIPMIP.

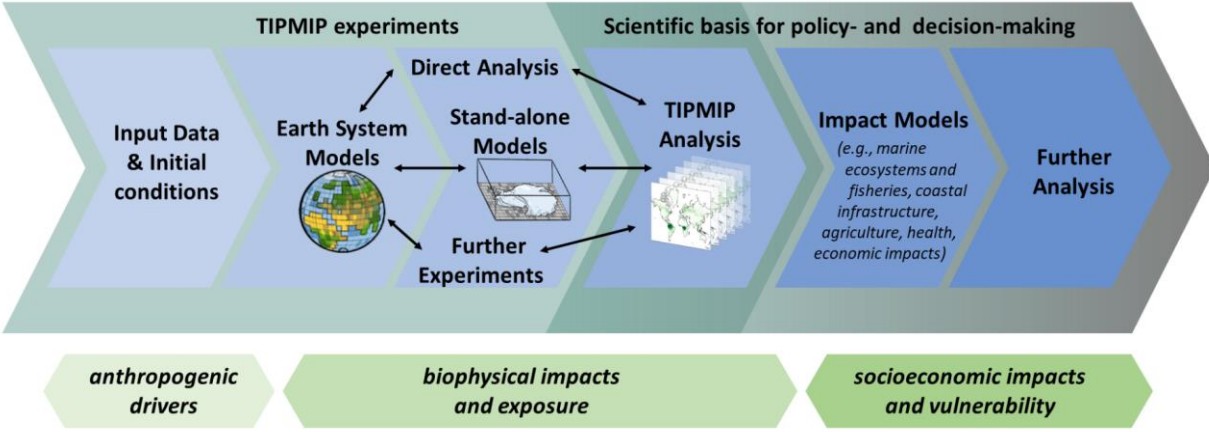

**Figure 8: TIPMIP workflow towards a more comprehensive scientific basis for policy- and decision-making.** Based on 645 input data and initial conditions, the initial set of TIPMIP experiments is run with coupled Earth System Models. The model output feeds directly into the overarching analysis, and is also used as input for extended simulations with stand-alone models (e.g., for the ice sheets where the time scales associated with committed impacts and tipping dynamics are generally longer)



as well as further experiments such as the 'What-if' - type simulations. The TIPMIP analysis brings the different types of experiments together to assess the biophysical impacts and exposure resulting from potential tipping dynamics in the Earth

system. TIPMIP output can further provide the basis for impact models to analyse effects on, e.g., agriculture, marine ecosystems and fisheries, coastal infrastructure, health, and economic impacts. Together, these results will generate a first global atlas of tipping dynamics and potential associated risks to inform policy- and decision-making.

The aim is to provide a building block towards a comprehensive scientific basis for policy- and decision-making (Fig. 8), not least in preparation of the upcoming IPCC assessment cycle (IPCC-AR7) and Global Stocktake as well as IPBES and the Earth Commission. We therefore call on the scientific community to support the model intercomparison and contribute via model simulations, comparison with observations, or analysis of the simulations. Participation in TIPMIP is open to all Earth system modelers and modelers of individual Earth system components as laid out in the domain description. Interested modelling

groups can contact the core team as detailed on the TIPMIP website (www.tipmip.org).

In addition, we envision the results to provide a basis for impact models to study the possible consequences and risks of nonlinear transitions in Earth system dynamics for human societies and the biosphere (e.g., within ISIMIP, Frieler et al., 2017), for instance with respect to agriculture, biomes/forestry, marine ecosystems and fisheries, coastal infrastructures, energy, and health, both at the global and (where possible) regional scale.

As the timeline for the upcoming high-level synthesis reports is quite ambitious, TIPMIP simulations will be conducted with current model configurations – these experiments in the first phase are aimed to also identify possible gaps with respect to certain feedbacks, and to give a first more systematic understanding of the respective uncertainties, which can then in turn inform model development and future experiments in a second phase of TIPMIP.

The next step after the domain-specific protocols (currently in preparation, based on several international protocol-design

workshops as well as domain meetings) are published and the respective input data is pre-processed (and, depending on the domain, bias-corrected), will be the simulation phase. The output data from all models will be submitted to the respective ESGF nodes and analysed by the modelling groups and associated TIPMIP scientists in a next step. The TIPMIP results will subsequently be opened for use by the wider scientific and stakeholder community, following the example of established model intercomparison projects such as CMIP or ISIMIP.

The development of this overarching description paper and the design of the experiments brought together scientists from very different disciplines and backgrounds with the joint goal of tackling some of the most critical and controversial questions in Earth system science today. We hope to further grow and establish this community and succeed in providing deeper and more advanced knowledge on possible risks through Earth system tipping dynamics and provide valuable information for the general public, stakeholders and policymakers.



**Code availability**

The python script used for generating Fig. 1 will be made available and archived in a public repository (zenodo). No further code was used in generating this article.

**Data availability**

No original data sets were used in this article.

**Author contributions**

Ricarda Winkelmann is the initiator and scientific lead of TIPMIP and conceived this study. All authors participated in the manuscript's design, writing and editing. Specific contributions were provided by:

- Writing of original draft: Ricarda Winkelmann, Donovan P. Dennis, Jonathan F. Donges, Sina Loriani, Ann Kristin Klose

- TIPMIP core team (coordination and administration): Ricarda Winkelmann, Donovan P. Dennis, Jonathan F. Donges, Sina Loriani, Karoline Ramin

- Domain leads and further domain-specific contributions (in alphabetical order):

  *TIPMIP-ESM:* Colin Jones, Torben Koenigk, Matteo Willeit, Shuting Yang, and Gokhan Danabasoglu

  *TIPMIP-ICE SHEETS*: Torsten Albrecht, Heiko Goelzer, Ann Kristin Klose, Marisa Montoya, Alexander Robinson, Robin S. Smith, Ricarda Winkelmann and Javier Blasco, Jorge Alvarez-Solas, Sophie Nowicki

  *TIPMIP-PERMAFROST:* Victor Brovkin, Eleanor Burke, Goran Georgievski, Jan Nitzbon, Christina Schädel, Norman J. Steinert

  *TIPMIP-OCEAN*: Aixue Hu, Laura C. Jackson, Didier Swingedouw, Wilbert Weijer, and Anastasia Romanou

  *TIPMIP-BIOSPHERE*: Anna B. Harper, Marina Hirota, Boris Sakschewski and Peter Lawrence

  *TIPMIP-REGIONAL*: Donovan P. Dennis, Hyungjun Kim, Bette Otto-Bliesner, Ricarda Winkelmann

  *TIPMIP-WHATIF*: Bette Otto-Bliesner, and Gabi Hegerl, Francesco Pausata, Steven Sherwood

  *TIPMIP-METHODS*: Jonathan F. Donges, Sina Loriani, Sebastian Bathiany, Richard Wood, and Reik V. Donner, José Licón-Saláiz, Klaus Wyser

- Further contributions (in alphabetical order): Jesse F. Abrams, David Armstrong McKay, Markus Drüke, Timothy M. Lenton, Hannah Liddy, Maxcence Menthon, Stefan Rahmstorf, Johan Rockström

- Visualisation: Sina Loriani, Karoline Ramin, Ricarda Winkelmann

**Competing interests**

Some authors are members of the editorial board of the journal Earth System Dynamics. Otherwise, the authors have no competing interests to declare.

680



**Acknowledgements**

TIPMIP received support from the Earth Commission. The work of the Earth Commission, a program of Future Earth, is made possible through the support of the Global Commons Alliance, a sponsored project of Rockefeller Philanthropy Advisors.

We acknowledge the financial support of the TipESM project funded by the European Union's Horizon Europe research and innovation programme under grant agreement No. 101137673: Funded by the European Union.

We further acknowledge the financial support by the ClimTip project funded by the European Union's Horizon Europe research and innovation programme under grant agreement No. 101137601: Funded by the European Union. This is ClimTip contribution #50.

This research was supported by Ocean Cryosphere Exchanges in ANtarctica: Impacts on Climate and the Earth system, OCEAN ICE, which is funded by the European Union, Horizon Europe Funding Programme for research and innovation under grant agreement No. 101060452, 10.3030/101060452. This is OCEAN ICE contribution #23.

This publication was supported by PROTECT. This project has received funding from the European Union's Horizon 2020 research and innovation programme under grant agreement No. 869304.

We acknowledge the financial support of the OptimESM project funded by the European Union's Horizon Europe research and innovation programme under grant agreement No.101081193: Funded by the European Union.

We acknowledge support from the European Research Council Advanced Grant project ERA (Earth Resilience in the Anthropocene, ERC-2016-ADG-743080).

*Views and opinions expressed are however those of the author(s) only and do not necessarily reflect those of the European Union or the European Climate, Infrastructure and Environment Executive Agency (CINEA). Neither the European Union nor the granting authority can be held responsible for them.*

J.F.D. acknowledges support by the project CHANGES funded by the German Federal Ministry for Education and Research (BMBF) under grant 01LS2001A.

C.S. received funding from the US Department of Energy (DOE) grant DE-SC0022116 and from Permafrost Pathways through the TED Audacious Project.

B.L O.-B. and G.H. acknowledge the Safe Landing Climates Lighthouse Activity of the World Climate Research Programme (WCRP) for supporting and facilitating the TIPMIP-WHATIF component of TIPMIP.

We thank all colleagues of the wider TIPMIP community, attending the various domain workshops, the WE-Heraeus Seminar on 'Addressing Key Uncertainties in Modelling Physical and Ecological Tipping Dynamics in the Earth System' in Templin (Germany, 2023) and the TIPMIP General Assembly, hosted by Future Earth/Earth Commission in Baltimore (US, 2024) for fruitful discussions of the experimental design.

685



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





## Appendix A

| Risk Transition | Global surface air temperature change (°C) | | Confidence |
|---|---|---|---|
| Undetectable to moderate | Min | 0.5 | *High* |
| | Max | 1.25 | |
| Moderate to high | Min | 1.25 | *Medium* |
| | Max | 1.75 | |
| High to very high | Min | 1.75 | *Low* |
| | Max | 2.25 | |

**Table A1:** This table is adapted from the supplementary material of Chapter 16 in the IPCC Sixth Assessment Report (AR6) Working Group II, which serves as the basis for the burning embers assessment of RFC5: Large-Scale Singular Events. The adaptation follows the assessment in Table S1 of the supplementary information in Rockström et al. (2023).

| Assessment year | Averaged global temperature anomaly [°C] |
|---|---|
| 2001 | 0.81 |
| 2009 | 0.92 |
| 2014 | 1.0 |
| 2018 | 1.12 |
| 2022 | 1.16 |
| 2023 | 1.45 |

**Table A2:** Global mean temperature values were averaged for each respective year using data from multiple observational and reanalysis datasets: HadCRUT5, NOAAGlobalTemp, GISTEMP, ERA5, JRA-55, and Berkeley Earth. The final values represent the mean across these datasets. Data sources were obtained from the Met Office Climate Dashboard, 2024.