# Peer review of "The Tipping Points Modelling Intercomparison Project (TIPMIP): Assessing tipping point risks in the Earth system"

_EGUsphere, 2025_

## Referee Comment (RC2)

**General comments**

This is the introductory paper for the Tipping Points Modelling Intercomparison Project. TIPMIP is an ambitious, multi-model international project that requires a substantial computational effort from participating ESM groups, more so than almost any other CMIP-sponsored MIP. The project is timely and important for informing climate policy decisions and future ESM development. Once published, the paper will be widely cited.

The manuscript is comprehensive and well organized. The authors have carefully thought through the project design, focusing on the systems most vulnerable to high-impact tipping events (ice sheets, permafrost, AMOC, and tropical and boreal forests) and proposing a set of coupled ESM experiments that, while ambitious, is streamlined enough to encourage participation.

The primary audience will be the modeling groups that would contribute to TIPMIP. These groups are under pressure to finalize their models and deliver DECK and scenario experiments for CMIP7. My understanding is that manuscripts need to be submitted by spring 2027 to be considered for the Working Group I report. Given this time pressure, ESM groups may be reluctant to add another set of experiments. With that in mind, I've suggested some changes to make the paper more accessible and persuasive to a broad audience.

This paper would be stronger if it were a bit shorter. Some passages are hard to read because of redundant information, complex sentence structure, use of passive voice, and long words where short words would do. Readers will have an easier time if the authors do some rewording and trimming. Also, many people won't read the entire paper, so it's helpful to present the main ideas before getting into too many scientific and technical details. Along these lines, the paper would benefit from some modest restructuring; see the suggestions below regarding Section 3.

I am concerned about the project timeframe. TIPMIP will inform the chapter on tipping points in AR7, which implies a tight timeframe for the experiments. I wonder if the manuscript was written before the WG1 submission deadlines were announced, and if knowing these deadlines has led to any reconsideration of scope. Is the AR7 timeframe practical for running all the first-phase experiments? If not, could the first phase could be divided into two parts, with the first part consisting of top-tier experiments that will inform AR7, and the second part including experiments that modelers could run later? Otherwise,

some groups might sit out the first phase of TIPMIP because there are too many experiments to run too soon.

I've divided my specific comments into two parts. The first part includes substantive comments, while the second is a list of wording suggestions that would make the paper easier to read. I don't expect the authors to adopt every wording suggestion, but I think they could adopt most of the suggestions without changing the meaning.

**Substantive comments**

l. 1 Here and below, the text often mentions "transgressing" critical thresholds. Since this word has unwanted connotations, please change to "crossing".

l. 22 The definition of tipping points refers to a change that is self-perpetuating, but self-perpetuating under what conditions? I suggest adding a few words to the effect of "when the external forcing is removed". Similarly at l. 114.

l. 51 See the general comment above on the project timeframe. This might be an appropriate place to describe the AR7 timeframe and how TIPMIP fits into that timeframe.

l. 79 "based on our current understanding tipping dynamics should in fact be considered 'unknown likelihood, high impact' events." Is this true? As the science improves, shouldn't we be able to assign likelihoods to some tipping events?

Fig. 2 The caption mentions the first phase of TIPMIP. Somewhere in Section 1, I suggest stating briefly what's included in the first phase (referring to Fig. 2) and what will be included in later phases.

Box 1 In general, the definitions here are clear and useful. But I have a question about abrupt changes: Why are these required to persist for at least a few decades and cause substantial impacts in human and/or natural systems? Wouldn't "abrupt" refer to the transition time to a new state, independent of persistence and impacts?

Fig. 3 Although the caption is clear, I'm not clear on the meaning of the arrows in the diagrams.

l. 163 "Due to the lifetime of $CO_2$ in the atmosphere, climate change can be considered irreversible on human timescales". This statement is questionable. How do we define a human timescale? Isn't it plausible that $CO_2$ could be drawn down on decadal timescales, especially with improved technologies?

Fig. 5   I had a hard time understanding this figure. It needs some more explanation to be accessible to many readers. Some familiar physical examples (e.g., ice sheet loss or AMOC collapse) might help explain the concepts.

l. 195   This is a long and technically detailed section. Some readers might give up at this point, before they get to the goals and structure of TIPMIP. I suggest putting this section after the current sections 4 and 5, which give the broad overview. You might even consider making it an appendix, since the scientific questions about specific systems are largely independent of the goals and structure.

l. 199   I think this is the first mention of the second phase. See the comment above on Fig. 2.

l. 365   Here you say, "Earth system models typically do not resolve most of those relevant vegetation dynamics." In the next paragraph you say, "we will assess the thresholds of tropical forests for large-scale tipping dynamics." A reader might ask how you can assess the thresholds if the relevant dynamics are missing. This is also a concert for other systems and processes.

I suggest that you address this concern in a general way, perhaps earlier in the section. For instance, suppose some critical process X is not included in any models. Is it still worth doing the experiments and analysis? The text partly answers this question by stating, "Lessons learned will then inform future experimental set-ups of ESMs participating in TIPMIP to include dynamic (rather than prescribed) vegetation changes." But some general comments near the beginning of this section would be helpful. Otherwise, TIPMIP could be criticized for trying to draw conclusions before the models are ready.

l. 446   "by integrating climate science, hydrology, disaster risk management, and socio-economic analysis".  This suggests that disaster risk management and socio-economic analysis are part of TIPMIP, instead of being beyond the scope. Please clarify (e.g., that TIPMIP modeling results will be a component of future integration that would involve these other disciplines).

l. 454   "we aim to generate an exposure risk assessment".  Again, this suggests an activity which is beyond the scope. I don't think "generate" is the right word here.

Box 2   This is a helpful summary of the key questions.

l. 464   "these key questions". It's not clear whether you mean the key questions in Box 2, or those mentioned in the previous paragraph. Since some of the questions in the previous paragraph are outside the scope, I assume you mean Box 2?

l. 469   commitment and stability timescales. The text doesn't define "long-term" with respect to the stability timescale, although Fig. 6 suggests a stability timescale of ~10 kyr.

That's a long time compared to typical ESM runs. Can TIPMIP really address all these timescales? Also, is it true that all systems would have the same commitment and stability scales? Won't some systems equilibrate and stabilize faster than others?

l. 486   "wealth of information". No doubt there will be large amounts of data, but modeling groups will be reluctant to allocate the required resources unless they think the effort is scientifically useful. It's clear that there are advantages to a common experimental design and that you will identify knowledge gaps. But you should try to make a stronger case that experiments which can be run with current models in 2026 will yield useful insights by 2027.

l. 493   I like the overall experimental design. Thinking about the AR7 timeframe: Is it possible to prioritize a subset of experiments that would be done in time to inform AR7?

l. 513   Some readers might have been thinking, until now, that this paper will describe protocols. I suggest that in Section 1, you say that more detailed information will appear in individual protocol papers, including Jones et al. for the ESM simulations.

l. 540   How does this project relate to ISMIP7, and how will you avoid overlap? Will the ESM output be ready in time to drive standalone ice sheet models on the AR7 timeframe? Do other domains (AMOC, etc.) require coordination with related MIPs?

l. 576   "overly (un)stable".  How would you assess whether the system is too stable or unstable, if the reason for excessive (in)stability is a missing process? Also, I was confused about "understand if and which simulated tipping dynamics are physically, biogeochemically or ecologically plausible". Are you referring to the physical plausibility of the interactions, or to the likelihood of the resulting tipping event?

l. 611   "The identified clusters of interest need to be scrutinised against conceivable positive feedback loops acting in those regions and system domains and against model characteristics such as hard-coded thresholds that can yield an abrupt shift wrongly associated with tipping dynamics." I'm not clear on the meaning here. I can see why hard-coded thresholds might lead to false tipping, but why would positive feedback loops be problematic?

l. 641   "More realistic scenarios are planned in a second phase". I had been thinking that the idealized forcing is an advantage. In the next phase, would it be better to repeat the idealized forcing experiments with improved models, as opposed to changing the scenarios? Of course, you can do both, but I'm wondering which would have greater value given limited resources.

l. 665   "As the timeline for the upcoming high-level synthesis reports is quite ambitious". Again, readers might be wondering if the first phase can be completed on this timeline.

***Wording suggestions***

l. 53  follows in -> follows

l. 54  "the need to further reduce the substantial associated uncertainties has remained high" -> "uncertainties have remained high"

l. 61  This list has no Oxford comma, but some lists use an Oxford comma. Please be consistent. I'd lean toward the Oxford comma to avoid ambiguity.

l. 69  Don't hyphenate "ice-sheets", here and elsewhere

l. 68  Delete "their self-sustained nature"

l. 71  Delete "it is clear that"

l. 74  Delete "very". This word appears often and in nearly all cases could be deleted.

l. 83  "to address the critical knowledge gaps regarding" -> "to improve our understanding of"

l. 93  Instead of "improve and consolidate", choose just one of these words. Likewise for "support and inform".

l. 95  Delete "into"

l. 96  Delete "today"

l. 96  "are confronted with" -> face

l. 97  "unprecedented climate impacts" -> "impacts"

l. 98  Delete "and could lead to unprecedented risks"; I think this goes without saying

l. 100  Delete "with respect to these aspects and characteristic traits"

l. 103  Delete "different types of"

l. 116  "for" -> "of"

Box 1  There is an extra carriage return in the first paragraph.

l. 122  "different kinds of spatial dynamics as well as different temporal dynamics" -> "different kinds of spatial and temporal dynamics"

l. 124  Delete "considerably"

l. 126  This is a long sentence that would be clearer if broken up.

l. 132  Delete "strongly"

l. 144  "a more appropriate extension of the instructive, but rather conceptual perspective of a simple double fold bifurcation is to distinguish" -> "it is useful to distinguish" or something similar

l. 160  "questions arise as to what this might entail in the long-term in terms of climate impacts, and especially their reversibility in context of different overshoot scenarios" -> "it is unknown whether many climate impacts are reversible in overshoot scenarios"

L. 194  "Significant ice loss from Antarctica and consequent sea-level rise for instance can be caused" -> "For instance, Antarctic ice loss can cause damaging sea-level rise …"

l. 195  Delete "critical"

l. 198  Don't capitalize "Semi-Arid" and "Arid"

l. 200  "This choice of focus systems is due to several reasons" -> "We focus on these systems because"

l. 201  "Transgressing critical thresholds even in parts of these core tipping elements" -> "Crossing tipping points"

l. 202  Delete "Moreover"

l. 206  "for the tipping systems considered here "-> "for these tipping systems"

l. 207  "across the model complexity hierarchy (Dijkstra, 2024) based on advanced understanding of the underlying biophysical processes (Brovkin et al., 2021; Boers et al., 2022), as summarized below". I think this sentence would get the main idea across if you deleted this text and just kept the references.

l. 208  "TIPMIP will be the first …". This has been said before.

l. 209  "In addition, we aim to study their interactions in coupled Earth system models, as the interconnected nature of the Earth system could lead to potential cascading effects (Wunderling et al., 2024), but little is known about such tipping cascades on a processdetailed level." This could be shorter, e.g. "We will study these systems in coupled Earth system models that can capture interactions and cascading effects."

l. 213   What does "prominent" mean here? E.g., the most studied, the most publicized, or those with the most severe impacts from tipping?

l. 226   Delete "mostly"

l. 227   "as ocean temperatures warm" -> "as the ocean warms"

l. 228   "cause an increase of" -> "increase"

l. 229   "are subject to" -> "rest on"

l. 231   "considerably" -> "much"

l. 236   I'm not sure you need "quasi" here.

l. 238   Delete "respective"

l. 239   Replace the colon with a period. This could be done in several other places.

l. 239   Run-on sentence. Maybe insert "where" before "recent studies"?

l. 241   "beyond the end of the century" -> "after 2100"

l. 242   "very few, individual modelling studies" -> "few modeling studies"

l. 247   "Next to" -> "Along with"

l. 257   Delete "that have been"

l. 276   This is a long sentence that could be broken up.

l. 283   Delete "prevailing"

l. 288   Delete "still"

l. 332   "represent" -> "are"

l. 336   Delete "also play a direct critical role in the Earth system, not least due to their potential to"

l. 341   "have already led to substantial changes in terms of biosphere integrity" -> "have already damaged biosphere integrity"

l. 345  "have been identified as" -> "are"

l. 350  "increase the likelihood of" -> "lead to"

l. 351  Delete "further"

l. 352  "Additionally" -> "Also"

l. 355  "significant uncertainties remain in predicting" -> "it is uncertain"

l. 356  "tropical forest tipping dynamics may occur" -> "tropical forests may tip"

l. 362  Delete "on strength and velocity of"

l. 365  Delete "relevant"

l. 374  "Too stable" is odd wording. I suggest deleting the clause in parentheses.

l. 379  "poses significant threats to" -> "threatens"

l. 384  "led to forest regeneration failures" -> "impeded forest regeneration"

l. 399  Delete "Immediate action in". It goes without saying that it's better to mitigate sooner than later.

l. 404  Delete "also"

l. 408  Start a new sentence after "dynamics"

l. 415  Delete "more"

l. 444  "critical thresholds in terrestrial hydrological systems will be assessed". Change to active voice ("we will assess").

l. 446  "this domain seeks to identify" -> "we seek to identify"?

l. 449  Delete "for TIPMIP"

l. 451  Delete "the characteristic traits and"

l. 466  "Ocean Models", etc. I suggest not capitalizing.

l. 468  "are aiming at addressing" -> "aim to address"

l. 473  "and/or interaction" -> "and interactions"

l. 480   Delete parentheses

l. 486   "we expect that a wealth of information will be gained from the TIPMIP experiments" -> "we expect the TIPMIP experiments to provide a wealth of information".

l. 488   Delete "key"; "prompt targeted" -> "guide"

l. 494   Delete "performed"

l. 498   Delete "respective", here and at l. 500

l. 501   "The reversibility of these changes and related timescales are assessed". Use active voice.

l. 529   "according domains" -> "eight domains"?

l. 533   Instead of "performs and analyses the experiments" in each paragraph, use this wording once, and then use something shorter like "focuses on".

l. 536   "to these are planned to" -> "will". Then you could add "explore" before "landing climates" and delete "to" before "increase" to make the structure parallel.

l. 602   Delete "prominent"

l. 606   Delete "ultimately"

l. 609   "introspection" doesn't seem like the right word here.

l. 616   What does "especially incipient" mean?

l. 621   Delete "corresponding"

l. 625   Delete "overarching description"

l. 635   Delete "in the future"

l. 639   "solidified manner" -> "consistent way"

l. 657   "call on" -> "invite". "Call on" implies an obligation rather than an act of good will.

l. 666   Change the dash to a period